# Targeting transcription factors through an IMiD independent zinc finger domain

Bee Hui Liu[1,16], Miao Liu [ID][2,16], Sridhar Radhakrishnan[1,16], Meng-Yuan Dai[3,16],

Chaitanya Kumar Jaladanki [ID][4,16], Chong Gao [ID][2,16], Jing Ping Tang[1], Kalpana Kumari [ID][1], Mei Lin Go[5],

Kim Anh L Vu[1], Junsu Kwon[2], Hyuk-Soo Seo [ID][6,7], Kijun Song [ID][6], Xi Tian[2], Li Feng[1], Justin L Tan [ID][1,8],

Arek V Melkonian[2,9], Zhaoji Liu[10], Gerburg Wulf[10], Haribabu Arthanari[6,11], Jun Qi[6,11], Sirano Dhe-Paganon[6,7],

John G Clohessy[12], Yeu Khai Choong[13], J Sivaraman[13], Hao Fan[4,14], Daniel G Tenen [ID][1,10,15✉] & Li Chai [ID][2✉]

## Abstract

**Immunomodulatory imide drugs (IMiDs) degrade specific C2H2 zinc finger degrons in transcription factors, making them effective against certain cancers. SALL4, a cancer driver, contains seven C2H2 zinc fingers in three clusters, including an IMiD degron in zinc finger cluster one (ZFC1). Surprisingly, IMiDs do not inhibit the growth of SALL4-expressing cancer cells. To overcome this limit, we focused on a non-IMiD domain, SALL4 zinc finger cluster four (ZFC4). By combining ZFC4-DNA crystal structure and an in silico docking algorithm, in conjunction with cell viability assays, we screened several chemical libraries against a potentially druggable binding pocket, leading to the discovery of SH6, a compound that selectively targets SALL4-expressing cancer cells. Mechanistic studies revealed that SH6 degrades SALL4 protein through the CUL4A/CRBN pathway, while deletion of ZFC4 abolished this activity. Moreover, SH6 treatment led to a significant 87% tumor growth inhibition of SALL4+ patient-derived xenografts and demonstrated good bioavailability in pharmacokinetic studies. In summary, these studies represent a new approach for IMiD independent drug discovery targeting C2H2 transcription factors such as SALL4 in cancer.**

**Keywords** SALL4; Degrader; IMiD; C2H2; Zinc Finger Cluster Four (ZFC4)
**Subject Category** Cancer

## Introduction

Transcription factors (TFs) have historically been viewed to be "undruggable" (Bushweller, 2019), with a number of attempts giving mixed results (Grembecka et al, 2007; Masso-Valles and Soucek, 2020; Rego et al, 2000; Verhoeven et al, 2020). Recent breakthroughs and successes in induced protein degradation are among the most exciting new frontiers for developing novel cancer drugs against TFs. Examples of such approaches include clinically approved Immunomodulatory imide drugs (IMiDs), or by proteolysis-targeting chimera (PROTAC) technology (Filippako-poulos et al, 2010; Jan et al, 2021). The IMiD drugs thalidomide, lenalidomide, and pomalidomide bind to cereblon (CRBN), a substrate receptor of the CUL4-RBX1-DDB1-CRBN (CRL4CRBN) E3 ubiquitin ligase, altering its substrate selectivity to recruit and degrade a specific set of proteins (Lu et al, 2014). Besides the known targeted proteins of IMiDs such as Ikaros (IKZF1), Aiolos (IKZF3), Casein kinase 1 alpha (CK1α), GSPT1, and SALL4, a set of Cys2-His2 (C2H2) zinc fingers have also been identified to be degraded by thalidomide analogs (Sievers et al, 2018).

SALL4 is a zinc finger transcription factor that is important for many cancers, most notably hepatocellular carcinoma (HCC) (Yong et al, 2013), lung cancer (Kobayashi et al, 2011a), ovarian cancer (Yang et al, 2016), breast cancer (Kobayashi et al, 2011b), myelodysplastic syndrome (MDS) (Liu et al, 2022), and acute myeloid leukemia (AML) (Li et al, 2013; Ma et al, 2006). While we have recently described a peptidomimetic molecule that has the ability to inhibit SALL4 function in HCC cell lines and xenograft mouse models (Liu et al, 2018), no small molecule targeted therapy against SALL4 in cancer is currently available.

SALL4 has 2 main isoforms. The larger isoform SALL4A is translated from the full-length SALL4 gene, while the smaller

[1]Cancer Science Institute of Singapore, Singapore 117599, Singapore. [2]Department of Pathology, Brigham & Women's Hospital, Harvard Medical School, Boston, MA 02115, USA. [3]Department of Oncology, Zhongnan Hospital of Wuhan University, Wuhan, China. [4]Bioinformatics Institute, Agency for Science, Technology and Research, Matrix, 138671 Singapore, Singapore. [5]Department of Pharmacy and Pharmaceutical Sciences, National University of Singapore, Singapore, Singapore. [6]Department of Cancer Biology, Dana-Farber Cancer Institute, Boston, MA 02215, USA. [7]Department of Biological Chemistry and Molecular Pharmacology, Harvard Medical School, Boston, MA 02115, USA. [8]Institute of Cancer Research, Shenzhen Bay Laboratory, Shenzhen 518132, China. [9]Wyss Institute for Biologically Inspired Engineering, Harvard University, MA, USA. [10]Department of Medicine, Beth Israel Deaconess Medical Center, Boston, MA 02115, USA. [11]Department of Medicine, Harvard Medical School, Boston, MA 02115, USA. [12]Preclinical Murine Pharmacogenetics Facility and Mouse Hospital, Department of Medicine, Beth Israel Deaconess Medical Center, Boston, MA, USA. [13]Department of Biological Sciences, National University of Singapore, Singapore, Singapore. [14]Cancer and Stem Cell Program, Duke-NUS Medical School, Singapore, Singapore. [15]Harvard Stem Cell Institute, Harvard Medical School, MA, USA. [16]These authors contributed equally: Bee Hui Liu, Miao Liu, Sridhar Radhakrishnan, Meng-Yuan Dai, Chaitanya Kumar Jaladanki, Chong Gao. ✉E-mail: dtenen@bidmc.harvard.edu; lchai@bwh.harvard.edu

isoform SALL4B is a spliced variant lacking the region of exon 2 encoding zinc finger clusters 1 and 2 (ZFC1 and ZFC2), while both isoforms contain ZFC4, the DNA binding domain (Kong et al, 2021). The crystal structure of SALL4 ZFC1, which is not present in SALL4B, has been shown to bind thalidomide (Furihata et al, 2020) and pomalidomide (Matyskiela et al, 2020) together with CRBN. We recently observed that although IMiDs are capable of degrading SALL4A through ZFC2, they do not target the SALL4B isoform. As a result, IMiDs do not diminish the viability of cancer cell lines that express both SALL4A and SALL4B (Kong et al, 2021; Vu et al, 2023). To effectively target SALL4+ cancer cells, both isoforms must be degraded, and to this end, we propose to target ZFC4, which is found in both SALL4A and SALL4B.

Previously, we and others have characterized the consensus DNA sequence to which SALL4 ZFC4 binds (Kong et al, 2021; Pantier et al, 2021). Here, we determined the crystal structure of the SALL4 ZFC4/DNA complex, and aided by AlphaFold (Bryant et al, 2022; Jumper and Hassabis, 2022), which predicts protein structure folds with high accuracy, we identified a potential binding pocket in SALL4 ZFC4. Screening of several chemical libraries for binding to this pocket led us to identify a small molecule lead compound SH6 which degraded SALL4 protein and inhibited the growth of SALL4+ cancer cells in culture and in vivo.

# Results

## Combining crystal structure, and molecular docking to screen known and novel drug libraries for SALL4 ZFC4 targeting compounds

SALL4 has two isoforms: SALL4A and SALL4B (Fig. 1A). We (Kong et al, 2021) and others (Pantier et al, 2021) independently found that the DNA binding domain of SALL4 is located in the fourth C2H2 zinc finger cluster (ZFC4) and that it prefers a consensus SALL4 DNA binding motif (Kong et al, 2021; Pantier et al, 2021). Unlike ZFC1, which is only present in SALL4A, ZFC4 is a common domain shared by SALL4A and SALL4B. To further determine the structural basis of SALL4-DNA interaction, we co-crystallized two molecules of SALL4 ZFC4 (amino acids 864–929) with a 12-mer blunt-end duplex DNA containing the SALL4 TATTA DNA binding motif (CGAAATATTAGC)(Kong et al, 2021). The crystal structure was solved at a resolution of 2.1 Å in a space group with an asymmetric unit that contained one molecule of the duplex DNA and two molecules of SALL4 ZFC4 (Fig. 1B(i)). The statistics are shown in Dataset EV1. While the first zinc finger of the second SALL4 molecule (ZFC4a') did not generate electron density, the remaining three zinc fingers (ZFC4a, ZFC4b and ZFC4b') generated high-quality electron densities, and bound to the duplex in canonical fashion, namely by inserting their helical elements into the major groove of DNA (Fig. EV1). Unlike the first finger ZFC4a, which provided only backbone interactions, the second finger (ZFC4b) interacted specifically with the amide side chain of Asn912 (N912), via the adenines at positions D4 and D5 of one strand and C10 of the other strand (Fig. EV1). Side chains of the second zinc finger of both molecules also provided four and five additional interactions with the DNA backbone, respectively, including Lys896-Y4, Arg905-Y3, Lys910-X4, Lys914-X5, and His916-Y2. Recently, the crystal structure of human SALL4 ZFC4

(856–930) (Ru et al, 2022) and murine SALL4 ZFC4 (870–940) (Watson et al, 2023) were published. All three structures were aligned in Fig. 1B(ii). Consistent with the report from Ru et al (Ru et al, 2022), our crystallographic findings of human SALL4 ZFC4 (864–929) bound to its consensus DNA binding target revealed an asymmetric unit that contained one molecule of the duplex DNA and two molecules of ZFC4. Both our crystal and the human ZFC4 (856–930) crystal (Ru et al, 2022) demonstrated that only one of the two zinc-fingers of the second ZFC4 molecule is bound. Furthermore, although all three structures contained different duplex DNA lengths and crystal packing, they displayed nearly identical overall conformations. Additionally, a close-up reveals the same key interactions, namely including hydrogen bonds between Asn912 and adenosine and a water-mediated interaction between Thr919 and thymidine, supporting a consensus among the structures.

We next tested the possibility of targeting SALL4 by molecular docking based on our knowledge of this newly characterized DNA binding structure. As the available ZFC4 crystal structure only encompasses a small region of the full-length SALL4 proteins, and there is currently no available SALL4B model in UniProt, we therefore used AlphaFold2 (Jumper et al, 2021; Jumper and Hassabis, 2022) to generate the 3D structural models of both isoforms (Fig. 1C). Our ZFC4 crystal data showed strong congruence with the AlphaFold models, with a root mean square deviation (RMSD) of 0.90 Å calculated using main chain atoms. We next predicted binding sites on the SALL4 isoforms using SiteMap (Schrödinger, 2020d). Docking studies were based on several criteria, including site area, volume, exposure, enclosure, contact properties, hydrophobicity, hydrophilicity, and donor-acceptor ratio. Sites with both a site score and Dscore (druggability) greater than 0.8 were prioritized for detailed analysis. For SALL4A, four binding sites were predicted: binding site 1 (BS1) in ZFC1 & 2; binding site 2 (BS2) in ZFC2 & 4; binding site 3 (BS3) within ZFC4; and binding site 4 (BS4) in ZFC2 (Fig. 1D). For SALL4B, only one binding site was predicted in ZFC4 that was identical to binding site 3 in SALL4A (SALL4B BS1, Fig. 1E). Not only is this site common to both SALL4A and B, it is also a site with potential clefts between the protein and DNA macromolecules that could be relevant for small molecule binding, including a water-filled albeit hydrophobic pocket of approximately 600 Å$^2$ formed at the intersection of the two zinc-finger molecules and the DNA. Therefore, this common site was chosen for subsequent work.

Chemical libraries with a total of 676 compounds from known (Sigma Aldrich LOPAC and Selleck Chem Anticancer Library) and novel (in-house SH Library) small molecule libraries were docked against SALL4A BS3 and SALL4B BS1 (Fig. 1F). Using an arbitrary cut off of −5.0 for docking scores, we shortlisted 119 compounds for further study (Datasets EV2, EV3). Interestingly, compounds from the in-house CSI-SH library displayed higher docking scores (mean docking score −6.8) as compared to the Selleck Chem Anticancer library (mean docking score-3.8) and Sigma Aldrich LOPAC library (mean docking score −4.1) (Datasets EV2, EV3), which duly prompted us to focus on the CSI-SH library. Structurally, the CSI-SH compounds are hybrid molecules derived from structural fragments of the microtubule inhibitor E7010 and the HDAC inhibitor N-(2-aminophenyl)-3-{4-[(phenylamino) methyl]phenyl}acrylamide, neither of which are known to interact with the SALL4 protein (Fig. 1G). The scaffold B-1 is constructed

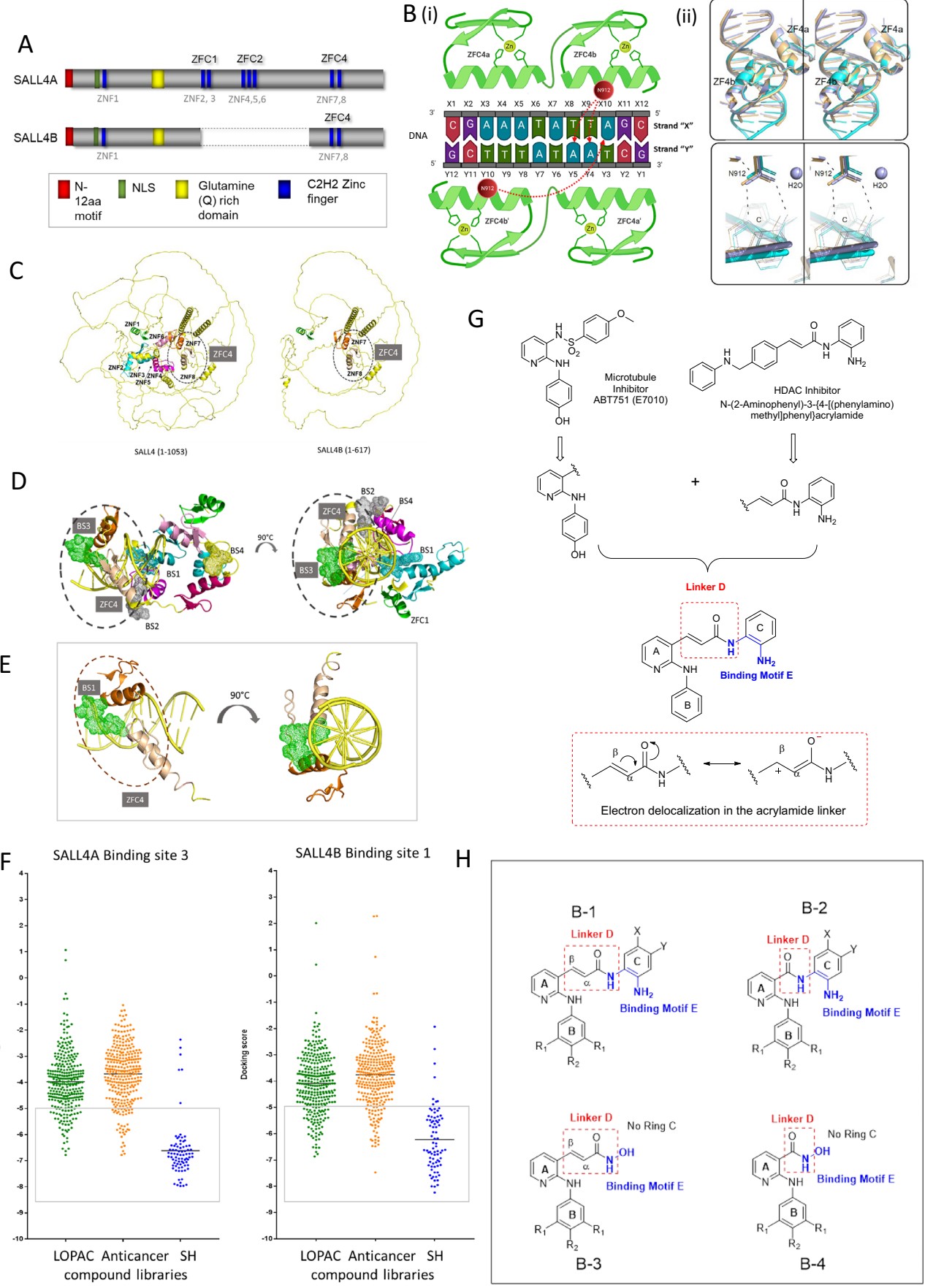

**Figure 1. Identification of a novel ZFC4 binding site.**

(A) Schematic representation of SALL4 protein isoforms. Dark blue rectangles = zinc fingers. SALL4A consists of four C2H2 zinc finger clusters (ZFC1, ZFC2, ZFC3, & ZFC4), while SALL4B, an internally spliced isoform, contains only ZFC1 and ZFC4. ZFC1 consists of Zinc Finger (ZNF) 1, ZFC2 consists of ZNF2&3, ZFC3 consists of ZNF4, 5, and 6, while ZFC4 consists of ZNF7&8. (B) Ligand Interaction Diagram of the SALL4 ZFC4-DNA complex is shown schematically. (i) Key SALL4 residues (N912) are labeled within red circles. Zinc ions are shown as small green circles. The duplex DNA strands are labeled arbitrarily with letters Y and X and numbered sequentially. Key interactions are shown as dotted red arrows. (ii) Alignment of our complex SALL4 structure (light blue) with recently deposited SALL4 complex structures 7Y3I (light orange) and 8A4I (cyan). (C) AlphaFold2 prediction of SALL4A and SALL4B. Left panel: ZNF1 (Green), ZNF2 (Cyan), ZNF3 (Teal), ZNF4 (magenta), ZNF5 (hot pink), ZNF6 (pink), ZNF7 (orange), and ZNF8 (wheat). Right panel: SALL4B, highlighted ZNF4 (ZNF7 (orange) and ZNF8 (wheat)). (D, E) Binding site prediction of SALL4 using SALL4 models from (A) and the ZFC4-DNA structure. Top view and side view of predicted binding sites on the SALL4A (D) and SALL4B (E) are shown. (F) Small-molecule docking screen against the common ZFC4 binding site in SALL4A (BS3) and SALL4B (BS1). Green = LOPAC1280 library, Orange = FDA approved anti-cancer library, Blue = SH library. Total number of compounds = 676. Shortlisted compounds (n = 119) with −5.0 cut off docking scores were highlighted in gray box, n = number of compounds. (G) The SH series of compounds are based on a hybrid design that incorporates the N-phenylpyridin-2-amino motif (rings A, B) of E7010 to the HDAC N-(2-aminophenyl) acrylamide moiety (ring C). (H) The following scaffolds are found in the SH series: B-1 in SH1-SH7, SH9-SH15, AH17-SH23, SH25-SH31, SH33-SH39 are characterized by an acrylamide linker D; B-2 is representative of SH41-SH47, SH49-SH55, SH57-SH63, SH65-SH71, SH73-SH79 in which linker E is an amide; B-3 are arylamides in which ring C is replaced by OH (SH8,SH16,SH24,SH32,SH40); B-4 are amides in which ring C is replaced by OH (SH48,SH56,SH64,SH72,SH80). Source data are available online for this figure.

by linking the (N-phenylpyridine-2-amino) moiety (rings A and B) of E7010 to the anilino ring C of the HDAC inhibitor via an acrylamide linker D (Fig. 1H). In another iteration of the scaffold, the acrylamide linker is replaced by an amide (B-2). Notable differences between the two linkers are length (acrylamide > amide) and extent of charge delocalization (acrylamide > amide) (Fig. 1G). Both scaffolds (B-1,B-2) retain the ortho-diamino group as a zinc binding motif E but this is replaced by hydroxamic acid (hydroxyl OH in place of ring C) in scaffolds B-3 and B-4 (Fig. 1H).

## SH6 as a SALL4 targeting compound

Cell-based screening was next employed to evaluate the growth inhibitory activity of the test compounds against SALL4 expressing malignant cells. SNU398, a liver cancer cell line with high endogenous levels of SALL4, and SNU387, a liver cancer cell line with undetectable levels of SALL4, were used for comparison. Cells were incubated with the test compound at 1 μM for 72 h, after which cell viability was assessed. Growth inhibition was observed for most of the test compounds, but only three members (SH2, SH6, SH7) displayed selective killing against SALL4 high cells (Fig. 2A). Next, we carried out dose escalation studies on all SH compounds to deduce their EC50 values on SNU387 and SNU398. In order to interrogate the effects of compounds targeting SALL4, we created isogenic cells by overexpressing SALL4A (SNU387-TgSALL4A) or SALL4B (SNU387-TgSALL4B) in SNU387 cells (Tan et al, 2019) (Fig. 2B, and Dataset EV4). The $EC_{50}$ of SH2, SH6, and SH7 in SALL4 high SNU398 cells were 0.4 μM, 0.5 μM, and 0.3 μM, respectively. In contrast, the $EC_{50}$ values of the same compounds were more than 10 μM on SALL4 low SNU387 cells. Interestingly, SH2, SH6, and SH7 were consistently more potent (by approximately 3-fold) against the SNU387-TgSALL4B (0.4 μM, 0.5 μM, and 0.3 μM, respectively) as compared to the SNU387-TgSALL4A (1.3 μM, 1.3 μM, and 1 μM, respectively).

Next, we compared the molecular properties of SH2, SH6, and SH7 (Fig. EV2A) using the cheminformatics tools Molinspiration (https://www.molinspiration.com/). It was found that lipophilicity or hydrophobicity increases in the order of SH7 > SH6 > SH2, and that SH6 is slightly more hydrophobic than SH2, with lesser solubility but higher permeability. We then determined the PAMPA permeability of SH6 and SH7 and found the effective permeability Pe of SH6 and SH7 to be $66.13 \times 10^6$ cm/s and $41.16 \times 10^6$ cm/s, respectively (Fig. EV2A).

When compared to controls carbamazine (highly permeable, carbamazine Pe = $65.22 \times 10^6$ cm/s) and antipyrine (poorly permeable, antipyrine Pe = $1.28 \times 10^6$ cm/s) (Fig. EV2B), SH6 and SH7 were deemed to have good permeability profiles, in keeping with their lipophilic character (mlog P 4.05 and 4.66, respectively). Due to its superiority in overall scores, we selected SH6 as the lead compound for further study.

To determine if SH6 could target other SALL4 high malignant cells, we tested SH6 against additional cancer cells expressing high level of SALL4 (H661, H838, CAL51) and cells expressing low level of SALL4 (H552 and H2030) (Fig. 2C). As before, a similar trend was again observed. We observed that SH6 displayed greater potencies against SALL4 expressing cancer cell lines H661, H838, and CAL51 (EC50 = 0.21 μM, 1.30 μM, and 0.54 μM, respectively) as compared to SALL4 low cell lines H552 and H2030 (EC50 = 48.4 μM, and 41.6 μM, respectively). Taken together, the results support the notion that SH6 preferentially targeted malignant cells that express high levels of SALL4.

We then examined the EC50 values of the SH compounds on SNU 398 (SALL4 high) cells for structure-activity correlations. Here, we noted three structural features associated with potent cell-based activity (Fig. 2D). First, the acrylamide linker D is a significant contributor to activity, exceeding that of the amide. Next is the ortho-diamino moiety, which is far superior to the alternative hydroxamic acid as a zinc binding motif (scaffolds B-3, B-4). Lastly, potent activity is associated with the bulky and lipophilic trimethoxylated ring B, possibly due to its involvement in reinforcing interactions at the target site. More details are provided in Fig. EV3. SH library compounds contain reactive aromatic acrylamide residues which could be potential Michael acceptors. To understand if SH compounds could form thiol-Michael adducts with nucleophilic residues in the biological milieu, we referred to the NMR spectroscopic method described by Avonto et al (Avonto et al, 2011) to identify thiol trapping agents. The ¹H NMR spectra of SH4, a close structural analog of SH6 with better solubility, was examined before and after 24 h incubation with the nucleophile cysteamine in deuterated DMSO (Appendix Fig. S1). Here we found no discernible difference in the spectra of SH4 and SH4+cysteamine. Notably, the SH4 trans double bond peaks (δ = 7.83 ppm and 6.82 ppm with J = 16 Hz) were observed in the SH4+cysteamine spectrum, suggesting the absence of adduct formation involving the acrylamide residue of SH4.

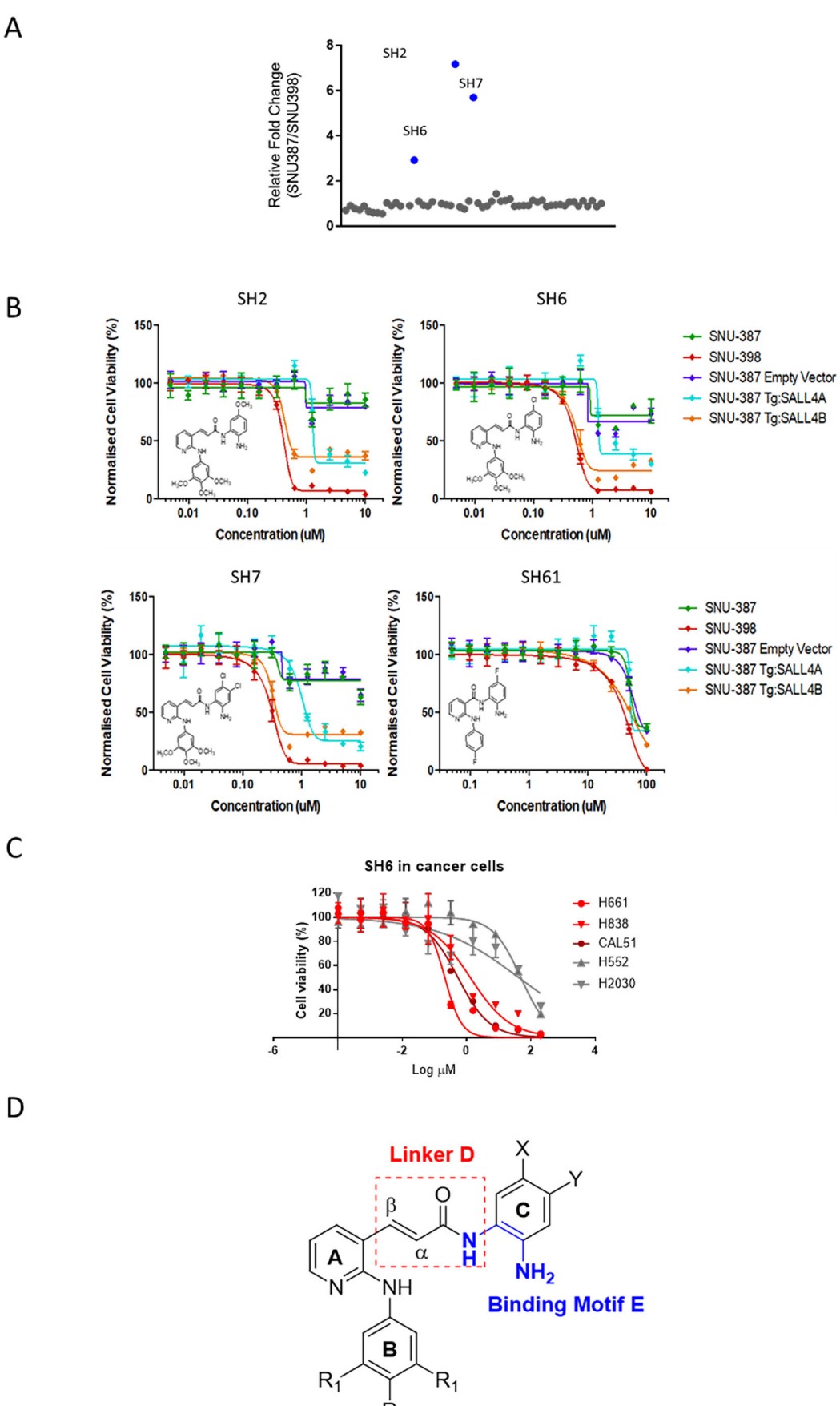

**Figure 2. Identification of a novel class of SALL4 inhibitor.**

(A) Single dose phenotypic screen using SALL4 high (SNU398) vs. low cell lines (SNU387) for 117 selected compounds from Fig. 1F, Datasets EV2, EV3; two of the compounds are not available for testing. (B) SH library compounds were tested for EC50 in cell viability assays. Isogenic cell lines specifically expressing SALL4A (SNU387-TgSALL4A) or SALL4B (SNU387-TgSALL4B) were tested alongside SNU398 and SNU387 cells. SH2, SH6, SH7, and SH61 are shown in the representative graphs. (C) SH6 was utilized in dose-dependent phenotypic screens using the SALL4 high lung cancer cell lines H661 and H838, as well as breast cancer line CAL51 (red) and SALL4 low lung cancer lines H1552 and H2030 (gray). (D) Schematic representation of three key structure-activity correlations in the SH library. A detailed description can be found in Fig. EV3. Each data point in (B) and (C) was performed in $n = 3$ independent cell cultures. IC$_{50}$ of compounds was determined using nonlinear regression fitting, with error bars representing the s.d. of the mean. Source data are available online for this figure.

## SH6 degrades SALL4B through the CUL4A-CRBN pathway

To understand the mechanism of SH6, we first probed SALL4A and SALL4B levels in SNU398 cells treated with SH6 or pomalidomide. While pomalidomide treatment led to a notable decrease in SALL4A protein, SH6 treatment led to decreases in both SALL4A and SALL4B isoforms (Fig. 3A). Intriguingly, in spite of its ability to degrade SALL4A, the IMiD lenalidomide failed to suppress the viability of SALL4 high SNU398 cells. In contrast, SH6 is able to selectively target SALL4 high cells (Fig. 3B). We recently found that depletion of SALL4B is required for targeting SALL4-mediated tumorigenesis (Vu et al, 2023). To further examine the effect of IMiDs and SH6 in degrading SALL4B, SALL4B was fused to NanoLuc (Lu et al, 2014) and stably transfected into H1299 cells. Cells were then treated with SH6 or lenalidomide for 24 h, followed by measurement of emitted luciferase as an indicator of SALL4B protein levels (Fig. 3C). The result demonstrated that SH6 treated cells emitted less luminescence as compared to lenalidomide treated cells, pointing to degradation of SALL4B by SH6 but not lenalidomide. Taken together, the results indicate that degradation of SALL4B by SH6 induced cell death, which could not be achieved by IMiDs in SALL4 expressing cancer cells.

Previous studies have reported that SALL4 is an IMiD-dependent CRL4$^{CRBN}$ substrate (Donovan et al, 2018; Filippako-poulos et al, 2010). This prompted us to interrogate the pathway involved in the IMiD independent, SH6-mediated SALL4 degradation. To this end, SNU398 cells were treated with either DMSO, the proteasome inhibitor MG132, the NEDD8-activating enzyme (NAE) inhibitor MLN4924, the IMiD thalidomide, a pan caspase inhibitor zVAD-FMK, or a lysosomal inhibitor chloroquine (CQ), prior to SH6 treatment. Cell lysates were harvested after 24 h of SH6 treatment, followed by western blot analysis (Fig. 3D). Interestingly, we noted that SH6 induced degradation of SALL4 was rescued by MG132, and more significantly by MLN4924, but not by zVAD or chloroquine. The result indicates that the SH6-SALL4 degradation pathway involves NEDD-8-mediated E3 ligase proteasomal degradation. When neddylation of cullin was blocked by MLN4924, SALL4 protein was rescued. Furthermore, when SNU398 cells were co-treated with MLN4924 and SH6, and cell viability was monitored over time (24 h, 48 h, 72 h), we observed a clear rescue profile at the latter two time points, which was noticeably absent in cells treated with SH6 alone under similar conditions (Fig. 3E). Taken together, these results implicate the involvement of an activated cullin pathway for SALL4 degradation.

We then employed a computation-guided model prediction approach to determine if CRBN-SALL4 interacts with CUL4A.

Using the existing crystal structures of the SALL4 ZFC1-CRBN-DDB1 complex as reference point, the crystal structures of the SALL4-CRBN-DDB1 complex (PDB ID: 6UML) and the DDB1-CUL4A-Rbx1 complex (PDB ID: 2HYE) were superimposed (with the predicted complex renamed as SALL4-CRL4$^{CRBN}$) (Fig. EV4A). When full-length SALL4 was superimposed on this SALL4-CRL4$^{CRBN}$ complex, we observed that ZFC1 and ZFC4 clash with CUL4A near the RBX1 domain, indicating that SALL4 and CUL4A might interact with each other. Next, protein–protein docking analysis was performed with CUL4A (extracted from PDB ID: 2HYE), SALL4A ZFC1 to 4, and SALL4B ZFC4. Thereafter, we superimposed the SALL4-CUL4A complexes with the SALL4-CRL4$^{CRBN}$ complex (defined above). We identified a potential binding site that appears in both ZFC1-4 of the SALL4A-CUL4A complex, and the ZFC4 of SALL4B-CUL4A complex (Fig. EV4B), coinciding with BS3 of SALL4A and BS1 of SALL4B identified previously (Fig. 1D,E). We docked SH6, SH61 (a negative control), as well as three thalidomide derivatives onto this binding site (Dataset EV5). Gratifyingly, SH6 outperformed the other five compounds in terms of docking score. With its docking pose wedged between ZFC4 of SALL4A/B and CUL4A, we hypothesized that SH6 may also bind to CUL4A (Fig. EV4B).

Next, we evaluated this model through a cell-based target engagement by SH6 using the cellular thermal shift assay (CETSA) (Jafari et al, 2014; Martinez Molina et al, 2013). Briefly, when a small molecule binds to a protein, it stabilizes the protein. Consequently, unfolding of the protein slows down at elevated temperatures. Here, we employed the in-cell CETSA to investigate the binding of SH6 to several components (NEDD8, CUL4, and CRBN) of the cullin pathway, and SALL4 itself. SNU398 cells were incubated with SH6 or DMSO for four hours prior to harvesting and lysis for CETSA analysis (Fig. 3F). We found that while the NEDD8 unfolding profile was not altered by SH6, the presence of SH6 led to higher thermal stability (less unfolding) of CUL4, CRBN, and SALL4 (Fig. 3F), implicating these proteins as binding partners of SH6. A similar CETSA was performed using the IMiD thalidomide, and only unfolding of CRBN, and not CUL4A, was delayed in the presence of thalidomide (Fig. 3G). Next, we examined the SH6-CUL4A binding using surface plasmon resonance (SPR). The affinity between the SH6 compound and CUL4A was reproducible at 3 different protein immobilization levels; KD was estimated at 89 μM, 98 μM, and 100 μM for low, medium, and high immobilization levels, respectively (Appendix Fig. S2). Taken together, our results indicate that even though SH6 is structurally different from IMiDs, it mediates SALL4 degradation via the CUL4A$^{CRBN}$ pathway. Unlike IMiDs, which do not directly target SALL4B, SH6 can degrade SALL4B, resulting in the death of SALL4 expressing cancer cells.

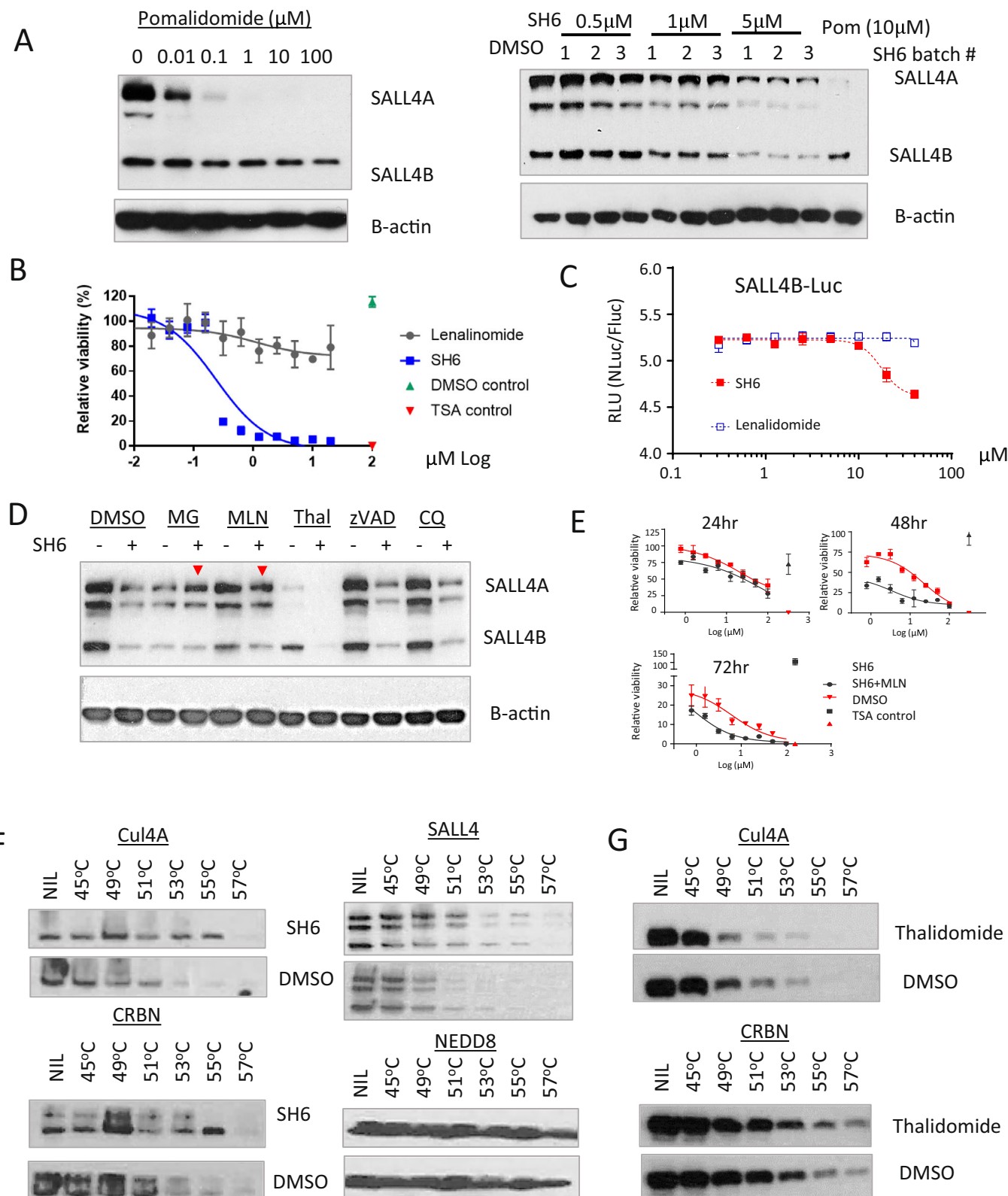

## SH6-mediated SALL4 degradation requires ZFC4

Our earlier findings highlight the significance of ZFC4 containing binding pocket in facilitating the degradation of SALL4B. To

further examine the importance of ZFC4 in SH6-mediated SALL4B degradation, we generated 293T cells that either expressed SALL4B (SALL4B^WT), or a SALL4B mutant lacking ZFC4 (SALL4B^-ZFC4). The cells were subsequently treated with SH6, lenalidomide, or DMSO

◄ **Figure 3.  IMiD independent SH6-mediated CUL4A-CRBN degradation of SALL4.**

(A) SNU398 cells were treated with pomalidomide or SH6 at the indicated concentrations for 24 h. Cells were harvested for western blot analysis. (B) Cells were treated with SH6 or lenalidomide for 72 h and examined for their viability. SH6 decreases cell viability while Lenalidomide does not show activity on these cells. (C) Dual-luciferase reporter assay. SALL4B was fused to the NanoLuc dual luciferase vector (pLL3.7-EF1a-IRES-Gateway-nluc-2xHA-IRES2-fluc-hCL1-P2A-Puro) and stably transfected into SALL4 + H1299 lung cancer cells. Cells were treated with SH6 or lenalidomide for 24 h and assayed for luminescence. DMSO normalized NLuc/FLuc data was presented as the mean ± s.d. of $n = 3$ biologically independent samples. (D) DMSO, MLN4924 (MLN), MG132 (MG), Thalidomide (Thal), zVAD-FMK (zVAD), or Chloroquine (CQ) were added for four hours prior to SH6 treatment, followed by western blot analysis 24 h later. MLN4924 and MG132 demonstrated prominent rescue of SH6 degradation of SALL4 (red arrows). Experiments were performed twice with independent cell cultures. (E) MLN4924 rescue of SH6 induced cell killing. SNU398 cells were incubated with SH6 or SH6 in combination with MLN4924 (SH6 + MN) for 24 h, 48 h, or 72 h. Cell viability was measured and data are presented as mean ± s.d. of $n = 3$ biologically independent samples. (F) Cells were incubated with SH6 for 4 h prior to CETSA analysis of the Cullin pathway. In-cell CETSA was performed as described in Materials and Methods. Cell lysates were separated by SDS-PAGE and analyzed by Western blotting. In-cell CETSA demonstrated that SH6 prolongs thermal stability for CUL4A, CRBN, and SALL4, in comparison to DMSO. (G) Cells treated with Thalidomide or DMSO underwent in-cell CETSA, revealing that Thalidomide stabilizes CRBN but not CUL4A. Each data point in (B), (C) and (E) was perform in $n = 3$ independent cell cultures. Inhibition curves were determined using nonlinear regression fitting, with error bars represents s.d. of mean. Source data are available online for this figure.

for 24 h, before harvesting for western blot analysis (Fig. 4A). The results indicate that SH6 effectively degraded wild-type SALL4B protein, but failed to degrade SALL4B in the absence of ZFC4. In contrast, lenalidomide did not induce degradation of either the wild type SALL4B or the SALL4B-ZFC4 mutant. Next, we performed a pull down assay using biotinylated SH6 in 293T-SALL4B[WT] and 293T-SALL4B[-ZFC4] cells. In agreement with the previous result, biotinylated SH6 pulled down intact SALL4B in 293T-SALL4B[WT] cells, but not in cells with mutant SALL4B lacking the ZFC4 domain (293T-SALL4B[-ZFC4]) (Fig. 4B). In addition, we performed in-cell CETSA in the 293T-SALL4B[WT] and 293T-SALL4B[-ZFC4] cells to investigate the binding of SH6 to SALL4 and CUL4A (Fig. 4C). Here, we observed that SH6 prolonged the stability of CUL4A in both wild type and mutant cells (Fig. 4C(i), (ii)). In contrast, SH6 only stabilized SALL4 in wild-type cells (Fig. 4C(iii)), and not in mutant cells without ZFC4, as compared to DMSO, indicating no interaction of SH6 with SALL4 in the mutant cells.

Next, we conducted a mass spectrometry analysis for additional SH6 targets. To do this, we treated SNU398 with SH6 for 24 h and then performed mass spectrometry with tandem mass tags. SALL4 was identified as the top protein downregulated by SH6, with 6-fold downregulation (Fig. 4D). Interestingly, all three top proteins (SALL4, ZFP91, and ZFP653) with more than 5-fold downregulation are C2H2 zinc finger-containing proteins. Other downregulated proteins (2-fold, $p$ values = 0.005) are listed in Dataset EV6. Using sequence alignments with Clustal Omega (Sievers and Higgins, 2018; Sievers et al, 2011), we found that the amino acid sequence of ZFP91 and ZFP653 showed high similarity (sequence identity 38% and 40%, respectively) and were similar to that of SALL4 ZFC4 (Fig. 4E). Based on these results, we conclude that SH6 is a genuine IMiD independent SALL4 degrader, and SH6-mediated SALL4 degradation requires ZFC4.

## SH6 inhibits tumor growth in mouse xenografts with a desirable pharmacokinetic profile

To determine if SH6 is selectively cytotoxic, we tested SH6 on two immortalized liver cell lines derived from normal liver (THLE2 and THLE3) (Fig. 5A). SH6 selectively kills SALL4 expressing liver cancer cells with EC$_{50}$ values of 0.3 μM to 1.3 μM (Dataset EV4); in comparison, SH6 demonstrated EC$_{50}$ values exceeded 25 μM against the two immortalized non-transformed THLE2 and THLE3 liver cell lines.

To further test the anti-tumor potential of SH6, SALL4-high SNU398 cells were implanted subcutaneously into the flanks of

NOD/SCID/Gamma mice (NSG), which were then randomly grouped for 20 mg/kg SH6 ($n = 20$), 40 mg/kg SH6 ($n = 13$) or vehicle control treatment ($n = 20$). Tumors in the vehicle control group progressively increased in size with mean tumor volume at 906.9 ± 137 mm$^3$ at harvest. In contrast, SH6 induced a strong therapeutic effect at 20 mg/kg and 40 mg/kg, with mean tumor volume at 604.4 ± 53 mm$^3$ and 345.9 ± 63 mm$^3$, respectively (Fig. 5B(i)). The body weight of mice was measured at the endpoint, and no significant change was observed between different treatment groups (Fig. 5B(ii)).

Next, we investigated the effect of SH6 in patient-derived xenografts (PDX) established from an HCC patient with a high level of endogenous SALL4 expression. In brief, patient tumor samples were surgically collected and screened for SALL4 expression, and subsequently implanted into NSG mice. A third-generation (F3) PDX passage from a 72-year-old male HCC patient with high SALL4 expression was used (Fig. EV5). Mice were randomly grouped and treated with either vehicle control ($n = 13$) or 50 mg/kg SH6 ($n = 13$) via intraperitoneal injection daily. Tumor volumes and body weights of mice were measured three times per week; treatment ended once the tumors in the control group reached the endpoint size of 1500 mm$^3$ in volume and the mice were euthanized (Fig. 5C(i)). The harvested tumors were photographed (Fig. 5C(i)), and analyzed with western blot for SALL4 expression (5 C(ii)). In this PDX cohort, tumors in the control group progressively increased in size, whereas tumor growth in the SH6-treated group was significantly inhibited (Fig. 5C(iii), (iv)). At the study endpoint, the mean tumor volume in the control group was 1495 ± 104.5 mm$^3$, compared to 186.3 ± 16.4 mm$^3$ in the SH6-treated group (Fig. 5C(iv). In this PDX study, SH6 induced a stronger therapeutic effect ($p < 0.0001$) with a tumor growth inhibition of 87%. To assess toxicity, we traced the body weight change throughout the experiment and found no significant change in either treatment group (Fig. 5C(v), (vi)). In addition, there were no differences observed in liver and renal function tests between the control and SH6-treated groups (Appendix Fig. S3).

To further assess the toxicity of SH6, we traced body weight change of 50 mg/kg SH6 treatment in four FVB mice for 12 days (Fig. 5D). The animals did not manifest behavioral changes and, interestingly, gained weight at the end of 12 days.

The in vitro microsomal stability and pharmacokinetic (PK) properties of SH6 were then investigated. Incubation with human liver microsomes revealed a half-life of 32.4 min, compared to 7.3 min for the rapidly cleared positive control verapamil (Fig. 5E).

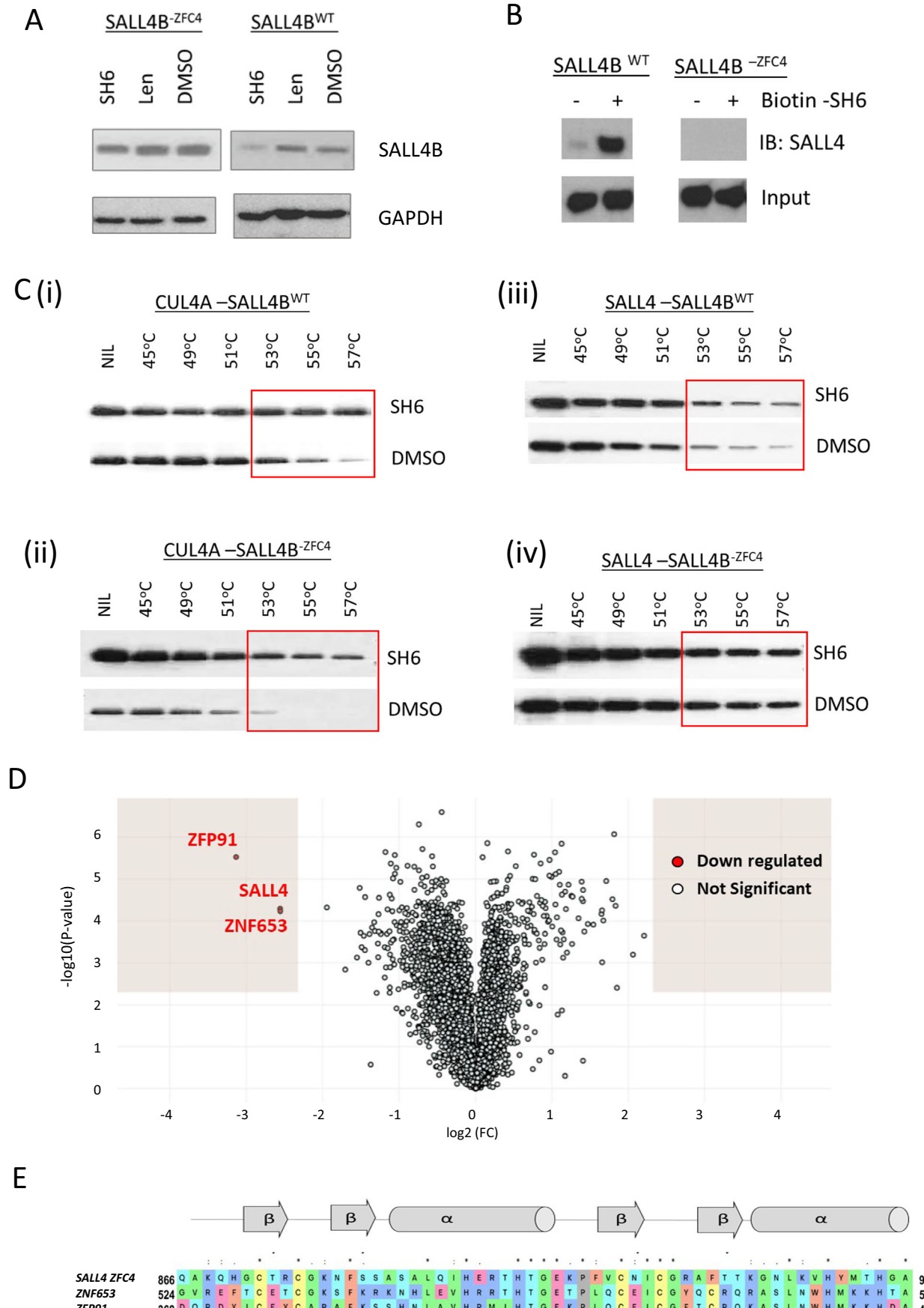

**Figure 4.  SH6 is a C2H2 ZFC4 degrader.**

(A) 293T cells were stably transfected with pFUW-SALL4B wild type (SALL4B$^{WT}$), or pFUW-SALL4B with deletion of ZFC4 (SALL4B$^{-ZFC4}$). Cells were treated with SH6, lenalidomide (Len), or DMSO for 24 h before harvested for western blot analysis. SH6-mediated degradation of SALL4B was abolished in the absence of ZFC4, while lenalidomide did not affect SALL4B levels in SALL4B$^{-ZFC4}$ or SALL4B$^{WT}$ cells. (B) Co-Immunoprecipitation pull down assay showing interaction of SALL4B and SH6. 293T SALL4B$^{WT}$ or SALL4B$^{-ZFC4}$ cells were treated with 2.5 μM biotinylated SH6 for 2 h, and bound proteins were eluted for western blot analysis. The assays demonstrated that SALL4B was bound and pulled down by biotinylated SH6 in SALL4B$^{WT}$ cells, but not in SALL4B$^{-ZFC4}$ cells. (C) 293T SALL4B$^{WT}$ or SALL4B$^{-ZFC4}$ cells were incubated with SH6 for 4 h prior to CETSA analysis. Cell lysates were separated by SDS-PAGE and analyzed by Western blotting for CUL4A and SALL4. The result demonstrated that SH6 prolonged thermal stability for CUL4A in both cell types ((i) and (ii)). SH6 also prolonged SALL4B stability in 293T SALL4B$^{WT}$ cells (iii), but not in SALL4B$^{-ZFC4}$ cells (iv), indicating no binding of SH6 to SALL4B without ZFC4. Red box highlighted the difference of the proteins under SH6 or DMSO treatment. (D) Proteomic study on SH6 treated SNU398 cells. A volcano plot from the mass spectrometry data demonstrating the change of the cellular proteins from the SH6 vs DMSO treatment. Downregulated proteins with twofold change and p-value of 0.05 were highlighted in red. (E) Sequence alignment of the SALL4 ZFC4 domain with ZFP91 and ZNF653. α-helices (α) and β-strands (β) are indicated according to data on crystal structures of SALL4 ZFC4. The color scheme used for residue conservation is based on side chain properties. Blue = positive charge (K, R, H); Cyan = polar (S, T, N, Q); Green=hydrophobic (A, I, L, M, V, G); Yellow = cysteine (C); Pink = negative (D, E); Orange = aromatic (F, Y, W). gray = proline (P). Source data are available online for this figure.

To further study the suitability of SH6 as a drug candidate, we assessed the blocking effects of SH6 on hERG potassium channels, which could cause life-threatening arrhythmias (Sanguinetti and Tristani-Firouzi, 2006) (Fig. 5F). SH6 and the positive control amitriptyline were investigated at concentrations of 0.3, 1, 3,10, and 30 μM. It was found that amitriptyline blocked hERG at an IC$_{50}$ of 2.76 μM, while SH6 blocked hERG at a much higher IC$_{50}$ of 22.2 μM.

Lastly, we evaluated the in vivo pharmacokinetic profile of SH6. Specifically, SH6 was administered to Swiss albino mice by intravenous injection (5 mg/kg) or oral gavage (15 mg/kg), after which blood samples were collected at specific time intervals (0.083, 0.25, 0.5, 1, 2, 4, 6, 8, and 24 h) for quantification of the drug by LC-MS/MS. As shown in Fig. 5G, SH6 has an acceptable PK profile characterized by good oral bioavailability (63.6%) and a reasonable half-life of 1.67 h. Importantly, it was well tolerated in the treated mice and no mortality was observed at the doses employed.

# Discussion

The ability to degrade a protein of interest opens up a plethora of possibilities in designing degraders. The development of JQ1 (Filippakopoulos et al, 2010), ER-PROTAC (Clinical trials NCT04072952), and AR-PROTAC (Clinical Trials NCT03888612), demonstrate the attention of the drug development community to the advantages of harnessing the E3-ubiquitin-proteosome pathway as a therapeutic target (Henley and Koehler, 2021). This strategy has recently been employed to degrade transcription factors (Choi et al, 2017; Verhoeven et al, 2020). As a transcription factor, the uniqueness of SALL4 lies in the fact that it is expressed in embryonic cells, downregulated during development, absent in most adult tissues, and re-expressed in a significant fraction of almost all human adult tumors (Moein et al, 2022; Tatetsu et al, 2016). The reappearance of SALL4 is associated with drug resistance in leukemia, lung, and breast cancer, and the onset of progenitor type hepatocellular carcinoma (Yong et al, 2013). The exclusive expression of SALL4 in cancer and not in adjacent untransformed cells provides an excellent therapeutic window that is rarely seen in drug targets. Immunomodulatory imide (IMiDs) drugs such as thalidomide have been shown to degrade SALL4 (Donovan et al, 2018; Matyskiela et al, 2018). It was reported that 5-hydroxythalidomide, a metabolite of thalidomide, targets the ZFC1 of SALL4 to Cereblon (Furihata et al, 2020). These studies explained, at least in part, thalidomide-induced birth defects, but failed to explain the lack of

potency of IMiDs in SALL4 expressing cancer cell lines. We have observed that IMiDs only degrade SALL4 isoform A and not SALL4 isoform B. Here, we used the newly characterized SALL4 DNA binding crystal structure, a domain shared by SALL4A and B, the artificial intelligence program AlphaFold2, as well as a cell-based assay to screen for compounds that only target cancer cells that are SALL4 positive. This strategy also helps to reduce cytotoxicity issues by maximizing the chance of a high therapeutic index. Using this screening approach, we identified the non-IMiD lead compound SH6, which degrades the SALL4 protein and selectively suppresses the viability of SALL4 expressing cancer cells. The AlphaFold 3D structure prediction tool provides confidence scores for each residue based on the local distance difference test score. These models have been useful in drug discovery (Flower and Hurley, 2021), identifying pathogenic mutations (Bryant et al, 2022), and to investigate protein–protein interactions (Porta-Pardo et al, 2022). Using a combination of crystallography of the ZFC4 DNA binding regions with AlphaFold2, we predicted and demonstrated that SH6 binding to SALL4A and SALL4B differed from the previously reported thalidomide binding site (i.e., cerebelon-SALL4A interface) (Furihata et al, 2020; Matyskiela et al, 2020), indicating that SH6 binds and functions differently from IMiDs.

In our study, we elected to employ the SALL4A and SALL4B AlphaFold models rather than the ZFC4:DNA crystal structure for the purposes of virtual screening, and this decision was predicated upon several critical considerations. While the ZFC4:DNA crystal structure shares a binding pocket region with the SALL4A/4B AlphaFold models, it is crucial to acknowledge that the ZFC4:DNA crystal structure only utilized a small segment of the protein. Specifically, it included only ZFC4, a mere 54 amino acids out of the 1053 amino acids in SALL4A or 616 amino acids in SALL4B. In contrast, the AlphaFold model utilizes the entire protein, including regions around the binding site that are not resolved in the ZFC4:DNA crystal structure, thereby affording a potentially more comprehensive representation of the molecular landscape for our virtual screening analysis. Furthermore, the AlphaFold models exhibit a complete binding site with minimal solvent accessible surface area, maintaining a more stable conformation that enhances the predictive accuracy of our molecular docking studies. In contrast, the binding site in the ZFC4:DNA crystal structure is highly exposed, making it more challenging to dock small molecules. The AlphaFold model predicted that SH6 could bind to SALL4, probably involving ZFC4. While the precise mechanism of SH6 binding to SALL4 remains to be determined, our studies do support that SH6-mediated SALL4 degradation requires ZFC4.

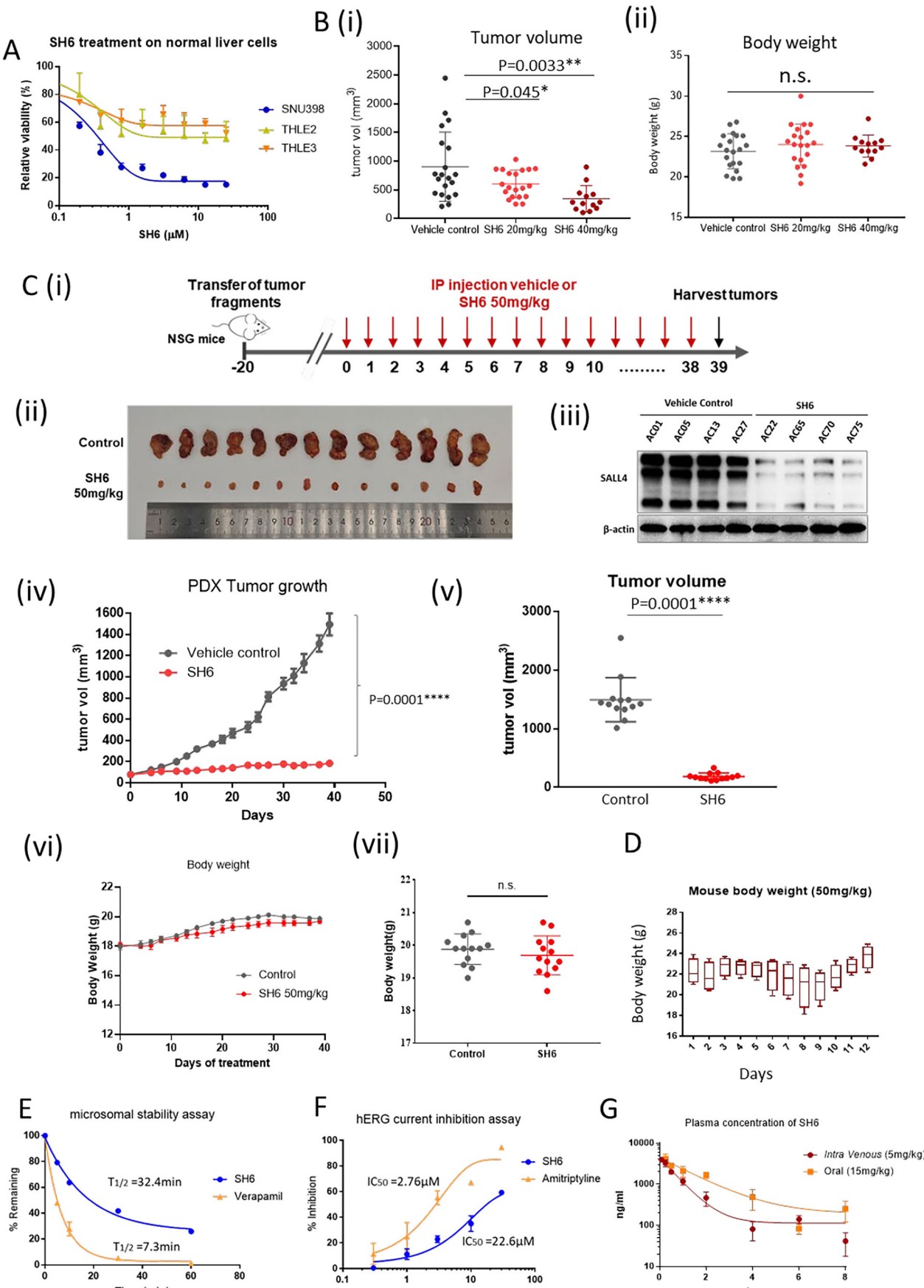

**Figure 5. SH6 leads to tumor regression and demonstrates a satisfactory pharmacokinetic profile.**

(A) SH6 specifically inhibits cell growth of SNU398 HCC but not non-transformed liver cell lines THLE-2 and THLE-3. (B) Mouse xenograft experiments. NSG mice were transplanted with $2 \times 10^6$ SNU398 cells, and the tumors were allowed to form a palpable mass. 20 mg/kg SH6 ($n = 20$), 40 mg/kg SH6 ($n = 13$) or vehicle control ($n = 20$) was administered every other day from Day 7 to Day 17. Tumors of the harvested xenografts at Day 17. Tumor weight (B (i)) and body weight of mice (B (ii)) were measured at end point. n.s. = not significant. P values were calculated by the two-tailed Student's t-test. Error bars = standard error of the mean (SEM). (C) (i) Patient-derived Xenograft (PDX) study design. Mice receiving PDX were randomly grouped and treated with vehicle control ($n = 13$) or 50 mg/kg SH6 ($n = 13$), $n =$ number of mice. Tumors harvested at end point were photographed C(ii). Four of the representative tumors from each treatment group were analyzed for SALL4 protein expression using western blot C(iii). Tumor growth was measured and traced ($n = 13$ for each group) (iv), and tumor volume measured at endpoint ($n = 13$ for each group) (v). N = number of mice (v). P values were calculated by the two-tailed Student's t-test. Error bars = standard error of the mean (SEM). Body weight of mice were also traced (vi) and measured at endpoint (vii). (D) Body weight of four healthy FVB/NJ mice receiving 50 mg/kg SH6 daily for 10 days. The plot represents the distribution of body weights across SH6 treated mice. The central line within each box corresponds to the median (50th percentile) body weight. The lower and upper bounds of the box represent the 25th (Q1) and 75th (Q3) percentiles, respectively, defining the interquartile range (IQR). Whiskers extend from the 25th percentile to the minimum value and from the 75th percentile to the maximum value, excluding any outliers. Error bars represent the standard error of the mean (SEM). All data are presented in grams (g). (E) Microsomal stability assay with 1 µM of SH6 or Verapamil. The percentage of the remaining compound (% Remaining, Y axis) was analyzed with LC-MS/MS. (F) Concentration response of hERG inhibition by SH6, with amitriptyline as the control. (G) Single dose pharmacokinetic study of SH6 via intravenous injection (5 mg/kg) or oral gavage (15 mg/kg) on Swiss albino mice ($n = 3$). The plasma concentration of the compound was measured with LC-MS/MS. Each data point in (A), (E), (F) and (G) was performed in $n = 3$ independent cell cultures. IC$_{50}$ of compounds in (A), (F) and (G) were determined using nonlinear regression fitting, with error bars representing the standard deviation of the mean. P values were calculated by the two-tailed Student's t-test. Source data are available online for this figure.

Cell-based screening was then employed to evaluate the selected compounds in targeting SALL4 expressing cancer cells. To ensure that the cell lines selected are representative and relevant for SALL4 study, we used cell lines with high and low endogenous SALL4 expression, as well isogenic cell lines overexpressing SALL4A or SALL4B (SNU387-TgSALL4A and SNU387-TgSALL4B) (Tan et al, 2019). This five-cell line cell-based screening platform has led to the discovery of the role of SALL4 in inhibition of oxidative phosphorylation and opens a new potential strategy to target SALL4 positive cancers (Tan et al, 2019; Yong et al, 2013). In this approach, SH6 showed selectivity against SALL4 high SNU398 cells when compared to SALL4 low SNU387 cells. Furthermore, it was found that SNU387-TgSALL4B were highly sensitive to SH6 treatment compared to SNU387 wild type and the SNU387-TgSALL4A isogenic line. Overall, our phenotypic screening results support that SH6 can selectively target cancer cell survival mediated by SALL4.

More recently, in other studies, the dependency on SALL4 in SNU398 cells was demonstrated through a shRNA-medicated knock down approach (Vu et al, 2023) in which we downregulated either total SALL4 (both A and B) or SALL4B only in SNU398 cells. We observed that upon SALL4/SALL4B-specific knockdown, the cellular growth ability of these cells was significantly inhibited, as tested by soft agar colony formation and clonogenic assays. In comparison, down-regulation of SALL4 had no effect on SNU387 cells (Vu et al, 2023).

An unforeseen discovery was made when we observed that SALL4 degradation caused by SH6 could be reversed with the use of proteasomal inhibitors MG132 and MLN4924. Despite the structural differences between SH6 and IMiDs, we did not anticipate the involvement of a similar E3 degradation pathway. Cullin-based E3 ligases need to be neddylated by NEDD8 in order to activate their holoenzyme ubiquitin ligase activities (Soucy et al, 2009). The neddylation itself is catalyzed by E1 ligase NAE, in which the formation of a NAE-NEDD8 thioester is crucial for subsequent transfer of NEDD8 to E2s and E3s. MLN4924 forms a covalent adduct with NEDD8 and inactivates it, blocking the very first step of NAE-NEDD8 formation. When SALL4 high cells are treated with MLN4924, SALL4 protein expression is rescued. Furthermore, the viability of SALL4 high cells, which is severely impaired by SH6, is rescued when co-treated with MLN4924 (Fig. 3E). These results indicate that SH6-mediated SALL4 degradation involves the cullin-based E3 system. Moreover, when we employed in-cell CETSA to investigate cell-based target engagement by SH6, we found that SH6 could engage

SALL4, CUL4A, and CRBN, causing significantly higher thermal stability for these proteins (Fig. 3F). A potential model has been proposed (see synopsis), in which IMiDs target ZFC1 of SALL4A, which is not present in SALL4B. This results in the inability of IMiDs to target SALL4+ cells which express both isoforms. In contrast, SH6 targets ZFC4, which is present in both SALL4A and SALL4B isoforms, and directs it to the CUL4-RBX1-DDB1-CRBN-ubiquitination-based degradation pathway, resulting in the death of SALL4+ cancer cells.

Herein, we developed a small molecule that degrades both SALL4 isoforms and showed prominent and specific efficacy in SALL4 expressing cancer cells. In our patient-derived xenograft (PDX) studies, SH6 achieved a remarkable 87% inhibition of tumor growth while allowing the mice to maintain weight gain and without impairing liver or renal function, underscoring its therapeutic safety profile. Additional cell line studies confirmed SH6's selectivity, targeting SALL4-positive cancer cells while sparing SALL4-negative cells and untransformed liver cells. Unlike traditional bulky PROTACs, SH6 combines a favorable pharmacokinetic profile with oral bioavailability, demonstrating exceptional antitumor efficacy in both murine PDX and xenograft models.

In summary, our study showcases the feasibility of combining crystallographic structural information of a zinc finger DNA binding transcription factor and AlphaFold2 (Jumper et al, 2021; Jumper and Hassabis, 2022), along with a phenotypic screen, to discover a targeted therapeutic molecule against the previously un-targetable oncofetal protein SALL4. We anticipate future medicinal chemistry studies to develop SH6 from a research tool compound to a clinically therapeutic drug targeting cancers with SALL4 expression.

# Methods

## Reagents and tools table

| Reagent/Resource | Reference or Source | Identifier or Catalog Number |
|---|---|---|
| **Experimental models** | | |
| FUW-Luc-mCh-puro-SALL4A | In-house | |
| FUW-Luc-mCh-puro-SALL4B | In-house | |
| FUW-Luc-mCh-puro | In-house | |

| Reagent/Resource | Reference or Source | Identifier or Catalog Number |
|---|---|---|
| NOD.Cg-Prkdc^scid Il2rg^tm1Wjl/SzJ (NSG) mice | Jax Laboratories | Stock# 00555 |
| FVB/NJ | In-house | |
| NOD-PrkdcscidIL2rgtm1 (NSG) mice | Beijing HFK Bioscience Co., Ltd. | Stock#14006A |
| **Recombinant DNA** | | |
| pLL3.7-EF1a-IRES-Gateway-nluc-2xHA-IRES2-fluc-hCL1-P2A-Puro-SALL4B | | |
| pLL3.7-EF1a-IRES-Gateway-nluc-2xHA-IRES2-fluc-hCL1-P2A-Puro | William Kaelin (Lu et al, 2014) | |
| **Antibodies** | | |
| SALL4 | Santa Cruz Biotechnology | sc-101147 |
| β-Actin | Santa Cruz Biotechnology | sc-47778 |
| Cul4A | Cell Signalling Technology | #2699 |
| CRBN | Cell Signalling Technology | D8H3S, #71810 |
| NEDD8 | ThermoFisher Scientific | PA517476 |
| secondary HRP-conjugated antibody to murine | Santa Cruz Biotechnology | sc-2005 |
| Armenian hamster IgG-HRP | Santa Cruz Biotechnology | sc-2789 |
| **Oligonucleotides and other sequence-based reagents** | | |
| **Chemicals, enzymes, and other reagents** | | |
| Nano-Glo®Dual-Luciferase® Reporter Assay System | Promega | N1650 |
| CellTiter-Glo | Promega | G7572 |
| Recombinant CUL4A | Creative Biomart | CUL4A-690H |
| **Software** | | |
| AlphaFold2 | Jumper et al, 2021; Jumper and Hassabis, 2022 | |
| SiteMap | Schrödinger, 2020d | |
| wizard module | Schrödinger, 2020c | |
| GLIDE module | Schrödinger, 2020a | |
| LigPrep module | Schrödinger, 2020b | |
| Piper | Kozakov et al, 2006 | |
| Prism | GraphPad | |
| **Other** | | |
| M1000 Microplate Reader | Tecan | |
| Biacore S200 | Cytiva | |
| Envision | Perkin Elmer | |

## Protein expression and purification

Residues 864–929 of human SALL4 were inserted into the pGEX4T1 (N-terminal GST thrombin tag) vector. A N-terminal

GST tagged construct of human SALL4 including residues 864–929 (RRQAKQHGCTRCGKNFSSASALQIHERTHTGEKPFVCNICGR AFTTKGNLKVHYMTHGANNNSARR, with SALL4 864–929 denoted in bold and underlined font) was overexpressed in *E. coli* BL21 (DE3) and purified using affinity chromatography and size-exclusion chromatography. Briefly, cells were grown at 37 °C in TB medium in the presence of 50 μg/ml of ampicillin to an OD of 0.8, cooled to 17 °C, induced with 500 μM isopropyl-1-thio-D-galacto-pyranoside (IPTG), incubated overnight at 17 °C, collected by centrifugation, and stored at −80 °C. Cell pellets were lysed in buffer A (25 mM HEPES, pH 7.5, 200 mM NaCl, 7 mM mercapto-ethanol, and 10 μM zinc chloride) using Microfluidizer (Micro-fluidics), and the resulting lysate was centrifuged at $30,000 \times g$ for 40 min. Glutathione superflow agarose beads (Fisher) were mixed with cleared lysate for 90 min and washed with buffer A. Beads were transferred to an FPLC-compatible column, and the bound protein was washed further with buffer A for 10 column volumes and eluted with buffer B (25 mM HEPES, pH 7.5, 200 mM NaCl, 7 mM mercapto-ethanol, 10 μM zinc chloride, and 15 mM glutathione). To cleave the GST-tag, thrombin was added to the eluted sample and incubated overnight at 4 °C. The resulting solution was concentrated and purified further using a Superdex 75 16/600 column (Cytiva) in buffer C containing 20 mM HEPES, pH 7.5, 200 mM NaCl, 0.5 mM TCEP, 1 mM DTT, and 10 μM zinc chloride. SALL4 containing fractions were pooled, concentrated to ~6 mg/mL, and stored in −80 °C.

## Crystallization

Using Formulatrix NT8 and ArtRobbins Phoenix liquid handlers, we pre-incubated 100 nl samples of 500 μM SALL4 with 750 μM 12-base pair blunt-end duplex DNA (CGAAATATTAGC) for 1 h. Subsequently, these samples were dispensed in an equal volume of crystallization buffer (comprising 30% PEG3350, 0.04 M $NH_4SO_4$, and BisTris at pH 6.0) and incubated against 25 μl of reservoir crystallization buffer in a 384-well hanging-drop vapor diffusion microtiter plate. The samples were incubated for three days at 20 °C and observed using Formulatrix Rock Imager. The crystals were briefly transferred into a crystallization buffer enriched with 25% glycerol before being flash-frozen in liquid nitrogen. They were then shipped to the synchrotron facility for data collection.

## Data collection and structure determination

Diffraction data were collected at beamline 24ID-E of the NE-CAT at the Advanced Photon Source (Argonne National Laboratory). Datasets were integrated and scaled using XDS (Kabsch, 2010). Structures were solved by SAD-phasing using the program SOLVE (Adams et al, 2010) Iterative manual model building and refinement using Phenix (Adams et al, 2010) and Coot (Emsley and Cowtan, 2004) led to a model with excellent statistics. Crystal structures, with statistics, shown in Dataset EV1, were deposited into the Protein Data Bank (PDB code 8CUC) with the PDB validation report.

## AlphaFold2

The 3D structural models of SALL4A and SALL4B were generated using AlphaFold2 (Jumper et al, 2021; Jumper and Hassabis, 2022)

through Colab python notebook (Matyskiela et al, 2018). Starting from the sequences of SALL4A and SALL4B, AlphaFold2 was run using "none" or "pdb70" template mode, MSA mode "mmseqs2 (Uniref + environmental), and pair mode "paired+unpaired" options. AlphaFold generated five structural models. The best ranked model by average pLDDT score was used for further analysis.

## Molecular docking prediction

Since AlphaFold models do not include ligands, SiteMap (Schrödinger, 2020d) (https://colab.research.google.com/github/sokrypton/ColabFold/blob/main/AlphaFold2.ipynb) was used to predict potential binding sites (Schrödinger, 2020a). To visualize and evaluate top binding sites, probable binding site regions with a minimum of fifteen site points were generated in a more restrictive hydrophobic environment using a standard grid. Based on Dscore and druggability score calculated by SiteMap, four pockets are considered for SALL4A and one pocket for SALL4B for further molecular docking analysis.

The AlphaFold models and crystal structure were prepared by a standard protocol protein preparation wizard module (Schrödinger, 2020c), which adds hydrogen atoms, repairs imperfect side chains and assigns protonated states of the system at pH 7.0 ± 2.0. The receptor grids for all proteins were generated on predicted binding sites from site map analysis using the GLIDE module (Schrödinger, 2020a) of Schrodinger. The generated site map points were used to generate a grid box 10 Å in size. Molecular docking was carried out using GLIDE module. Before docking, The LigPrep module (Schrödinger, 2020b) was utilized to prepare the ligands at pH = 7.5 ± 1, for tautomer generation, and energy minimization with the OPLS3e force field. Ligands were docked on predicted binding sites in the standard precision (SP) mode to generate 10 poses per ligand through flexible ligand sampling. Their binding poses were evaluated using Glide SP scoring functions.

The ZFC2-4 of SALL4A and ZFC4 of SALL4B models were docked to CUL4A using Piper (Kozakov et al, 2006). 70,000 rotations were sampled, and the top 30 poses were returned from each docking job. We considered the best scored poses for further analysis.

## Cell-based phenotypic screen

Empty vector, SALL4A, and SALL4B expressing isogenic cell lines were generated by transducing the SALL4 negative hepatocellular carcinoma SNU-387 cells with empty vector, SALL4A, or SALL4B FUW-Luc-mCh-puro lentiviral constructs. Cells were plated in 50 μl of RPMI culture media in 384-well white flat-bottom plates (Corning) and incubated at 37 °C in a humidified atmosphere of 5% $CO_2$ overnight. Cell numbers per well were 1500 for SNU-398, and 750 for SNU-387 and SNU-387 isogenic lines. After overnight incubation, varying concentrations of compounds 1–80 were added to cells with multichannel electronic pipettes (Rainin). Cells were then incubated for 72 h at 37 °C in a humidified atmosphere of 5% $CO_2$ before 10 μl of CellTiter-Glo reagent was added to the wells with the MultiFlo Microplate Dispenser (BioTek). Cells were incubated at room temperature for a minimum of 10 min, after which luminescence readings were recorded by an Infinite M1000 Microplate Reader (Tecan). Each data point was performed in $n = 3$

independent cell cultures. $IC_{50}$ of compounds were determined using nonlinear regression fitting from Prism (GraphPad).

## Western blots

SNU398 high SALL4 HCC cells were incubated with the compounds (SH6 or pomalidomide) for concentration and times indicated in the figure legends. Cells were then harvested by cell scraper and washed with PBS. The collected pellets were lysed with RIPA buffer (50 mM Tris, 150 mM NaCl, 1% Triton X-100, 0.5% sodium deoxycholate, and 0.1% SDS) supplemented with Complete™ protease inhibitor cocktail (Roche, Switzerland). The extracted protein lysates were denatured with 4X SDS sample buffer (200 mM Tris-HCl pH 6.8, 8% SDS, 40% glycerol, 4% β-mercaptoethanol, 50 mM EDTA, 0.08% bromophenol blue) at 99 °C for 5 min. Equal amounts of protein were subjected to electrophoresis in 8% SDS-PAGE gels, and then transferred to PVDF membranes. After blocking in Blocking One (Nacalai Tesque), the membrane was probed with primary antibodies to SALL4 (Santa Cruz Biotechnology, sc-101147), β-Actin (Santa Cruz Biotechnology, sc-47778), Cul4A (#2699) Cell Signalling Technology, CRBN (D8H3S, #71810) Cell Signalling Technology.

After washing with TBS-T, membranes were incubated with secondary HRP-conjugated antibody to murine (Santa Cruz Biotechnology, sc-2005), or Armenian hamster IgG-HRP (Santa Cruz Biotechnology, sc-2789) for 1 h. Luminata™ western HRP substrate (Millipore) was applied to the membrane for visualization.

## Dual-luciferase reporter assay

SALL4B was subcloned into Nluc/Fluc plasmid (pLL3.7-EF1a-IRES-Gateway-nluc-2xHA-IRES2-fluc-hCL1-P2A-Puro) provided by William Kaelin (Lu et al, 2014) and stably transfected into H1299 cells (H1299-SALL4BNLuc). H1299-SALL4BNLuc cells were seeded at 1000 cells per well in 384-well plates, and treated with drugs for 24 h before dual luciferase assays were performed using the Nano-Glo®Dual-Luciferase® Reporter Assay System (Promega, Cat#: N1650) as described by the manufacturer. Luminescence was read by Envision™ (Perkin Elmer) after 24 h of drug treatment. Data was presented by comparing DMSO normalized Nanoluc luminescence (NLuc) to DMSO normalized Firefly luminescence (FLuc). Biological triplicates were performed for each drug concentration. Each data point was performed in $n = 3$ independent cell cultures. $IC_{50}$ of compounds were determined using nonlinear regression fitting from Prism (GraphPad).

## Cell culture

Cell lines ((SNU398 (CRL-2233), SNU387 (CRL-2237), THLE2 (CRL-2706), THLE3 (CRL-3583), A549 (CRM-CCL-185), H1299 (CRL-5803), H838 (CRL-5844), H552 (CRL-5810), H2030 (CRL-5914), and H661 (HTB-183)) were obtained from ATCC and grown according to the provider's instructions in the absence of antibiotics. CA51 (CVCL_1110) was obtained from DSMZ (Leibniz Institute DSMZ-German Collection of Microorganisms and Cell Cultures GmbH) and grown according to provider's instructions in the absence of antibiotics. Cell lines were maintained in Dulbecco's modified Eagle's medium (DMEM) and Roswell Park Memorial Institute 1640 medium (RPMI) (Life Technologies, Carlsbad, CA)

with 10% fetal bovine serum (FBS) (Invitrogen) and 2 mM L-Glutamine (Invitrogen). These cell lines were cultured at 37 °C in a humidified incubator with 5% $CO_2$. Cell line authentication were performed by the supplier and cells tested negative for mycoplasma.

## Cellular thermal shift assay (CETSA)

Cells were treated with 50 µM SH6 or DMSO for 4 h in 37 °C with 5% of $CO_2$. Cells were harvested and lysed in 1X NP-40 lysis buffer containing 10 mM Tris-HCl (pH 7.4), 150 mM NaCl, 2.7 mM KCl, and 0.4% NP-40. The mixture was then rotated end-over-end for 30 min at 4 °C for lysis. The lysate was separated from the cell debris by centrifugation at 20,000 rpm for 10 min at 4 °C. Lysates were heated in a thermal cycler at different temperatures, ranging from 45 °C to 57 °C for 3 min, followed by a 3-min cool-down at room temperature. The heated lysates were centrifuged at 15,000 rpm for 10 min at 4 °C to separate the precipitates from the soluble fractions. The supernatants were subjected to western blot analysis with CUL4A (Cell Signalling Technology-2699), NEDD8 (ThermoFisher Scientific-PA517476), SALL4 (Santa Cruz Biotechnology, sc-101147), and CRBN (Cell Signalling Technology-71810).

## Surface plasmon resonance

The experiments were performed on a Biacore S200 Instrument (Cytiva) at 25 °C. The immobilization buffer was 50 mM Hepes; 200 mM NaCl; 0.005% Tween 20; 0.2 mM TCEP; supplemented with 1% DMSO when testing the interactions between SH6 compound and the immobilized protein. A sensor chip CMD 200 M (Xantec) was used for covalent immobilization of CUL4A (Creative Biomart CUL4A-690H) via amine coupling (Amine coupling kit, Cytiva). Following the manufacturer's recommendations, the surface was activated by EDC/NHS for 420 s at 5 µL/min. The protein was diluted in 10 mM acetate buffer pH 5 and injected on the surface at 5 µL/min until reaching 1700 RU; 2800 RU; and 4000 RU on Fc 2; 3; and 4. The surface was deactivated by injecting ethanolamine for 420 s at 5 µL/min. Fc 1 was used as a reference: the surface was activated by EDC/NHS and deactivated by ethanolamine. The interaction between SH6 compound and immobilized CUL4A was tested by multiple cycle kinetics (MCK): the compound was injected on all channels at a range of concentrations using 90 s association, 120 s dissociation at 50 µL/min. Results were analyzed using Biacore S200 software. Dissociation constants were estimated using the Langmuir 1:1 interaction model on double-subtracted sensorgrams. Steady-state analysis was not performed due to the absence of saturation.

## Mass spec analysis

In brief, SNU398 was grown treated with SH6 (2.5 µM) or DMSO in triplicates for 24 h. 100 µg of each sample's cell pellet was collected and submitted to the Thermo Fisher Center for Multiplexed Proteomics (TCMP) at Harvard Medical School. 25 µg of protein from each sample was reduced with Tris(2carboxyethyl)phosphine, alkylated with iodoacetamide, and then further reduced with DTT. Proteins were precipitated onto SP3 beads to facilitate a buffer exchange into digestion buffer. Samples were digested with Lys-C (1:50) overnight at room temperature and trypsin (1:50) for 6 h at 37 °C. Peptides were labeled with TMTPro reagents. 2 µL of each sample was pooled and used to shoot a ratio check in order to confirm complete TMT labeling. All 18 TMTPro-labeled samples from each group were pooled and desalted by Sep-pak. Peptides (whole proteome) were fractionated into 24 fractions using basic reverse

phase HPLC. 12 fractions were solubilized, desalted by stage tip, and analyzed on an Orbitrap Lumos mass spectrometer. MS2 spectra were searched using the COMET algorithm against a Human Uniprot composite database containing its reversed complement and known contaminants. For proteome, peptide spectral matches were filtered to a 1% false discovery rate (FDR) using the target-decoy strategy combined with linear discriminant analysis. The proteins were filtered to a < 1% FDR and quantified only from peptides with a summed SN threshold of >180. The Excel data sheet contains quantified data as "summed intensity" and "normalized relative abundance" (%). "Summed intensity" is the abundance value of all the peptides. "Normalized relative abundance (%)" is the abundance value that is multiplied by a factor so that the sum of the abundances across all proteins is equal. Data are available via ProteomeXchange with identifier PXD061151.

## HCC mouse xenografts and toxicity study

All animal studies were carried out with approval from the BIDMC Institutional Animal Care and Use Committee (IACUC) under protocol numbers 077-2021 and 052-2020-23. All mice were housed in Allentown ventilated cages with 60 air changes per hour and vivarium rooms were maintained at 70–72 °F with between 40 and 60% humidity. For HCC xenograft study, female NOD.Cg-Prkdc^scid Il2rg^tm1Wjl/SzJ (NSG) mice (Stock# 00555, Jax Laboratories) of 6–8 weeks of age were shaved on the right flank and injected subcutaneously with $5 \times 10^5$ SNU398 cells in a volume of 200 µL (1:1 serum-free media and Matrigel (BD Biosciences)). Tumors were allowed to engraft for four days and were then enrolled on treatment. Mice were treated daily with vehicle, 20 mg/kg, or 40 mg/kg SH6 prepared in 10% ethanol:10% Cremophor:70% phosphate buffered saline (PBS). Tumor volumes were estimated using the formula volume = ½(length × width²) and tumors were measured twice weekly using digital calipers. Mice were euthanized 19 days after enrollment on vehicle or SH6, body weights taken, and tumors harvested and weighed. SH6 toxicity studies were carried out in four healthy FVB/NJ female mice. The treatment group of mice received an i.p. injection of 50 mg/kg body weight (BW) of SH6 daily for 10 days in the lower right quadrant of the abdomen. Mouse body weight (BW) was measured every day during the treatment. No blinding was done.

## Patient-derived xenograft (PDX) model

Patient-derived xenograft (PDX) studies were conducted at the Model Animal Research Institute of Wuhan University. Fresh liver cancer tissues were obtained from liver cancer patients at Zhongnan Hospital of Wuhan University who were pathologically diagnosed with hepatocellular carcinoma (HCC). All clinical samples were obtained with approval by the Ethics Committee of Zhongnan Hospital of Wuhan University (No. 2022076K), and informed consent was obtained from each patient. Tumor samples were collected during surgery, rinsed 2–3 times in pre-cooled saline solution containing penicillin, kanamycin sulfate, and amphotericin B, and the non-tumor tissue was trimmed away. Part of the tissue was preserved for Western Blot analysis to assess SALL4 expression. Tumor tissues were then cut into small tumor pieces of 1–2 mm³ and placed in pre-cooled DMEM medium for further use.

Male NOD-PrkdcscidIL2rgtm1 (NSG) mice (Stock#14006A, Beijing HFK Bioscience Co., Ltd.) aged 4–6 weeks were used for experiments. NSG mice were bred and/or maintained in a specific-

pathogen-free animal facility at 22–26 °C and 40–60% humidity under 12/12-h light/dark cycle and up to five animals per cage, with access to food and water ad libitum. An incision was made near the base of the hind limbs of the NSG mice, and a trocar was used to insert the small tumor pieces into the subcutaneous region of the mouse. After placing the tumor pieces, the needle was withdrawn and the incision site was disinfected with iodine. Tumor proliferation was monitored daily, and when tumors reached a size of 400–800 mm³, they were harvested from the host and passaged into new NSG mice. For subsequent experiments, a third-generation PDX model derived from a 72-year-old male HCC patient with high SALL4 expression was used. When tumor sizes reached 80 mm³, the mice were randomly divided into a control group and a treatment group. Mice were treated daily with either the control solvent or SH6 (50 mg/kg, intraperitoneal injection). Tumor volume and body weight were monitored three times per week. Mice were euthanized and tumors collected when tumor diameters in the control group exceeded 1500 mm³. The tumors were then flash-frozen and stored at −80 °C for future use. All clinical samples were obtained with approval from the Ethics Committee of Zhongnan Hospital of Wuhan University (No. 2022076K), and informed consent was obtained from each patient. No blinding was done.

## Synthesis of SH6

(i) NaOBu-t, Pd(OAc)$_2$, PPh$_3$, xylene, 3 h; (ii) tert-butyl acrylate, Pd$_2$(dba)$_3$, DMP, DiPEA, 120 °C; (iii) TFA, DCM, RT; (iv) 4-chloro-1,2-phenylenediamine, HATU, DiPEA, DMF, RT.

Synthesis of 2, 3-Bromo-N-(3,4,5-trimethoxyphenyl)pyridine-2-amine (**2**): In a 5-ml screw-capped vial, palladium acetate (0.14 mg, 0.62 mmol), 4,5-bis(diphenylphosphino)-9,9-dimethylxanthene (Xantphos) (0.73 g, 1.26 mmol), cesium carbonate (8.25 g, 25.2 mmol), 2,3-dibromopyridine (**1**, 3.0 g, 12.6 mmol) and degassed toluene (30 ml) were mixed, and to this, trimethoxyaniline (2.31 g, MW-183, 12.6 mmol) was added. The reaction vial was flushed with nitrogen gas, stoppered tightly and the mixture stirred with heating at an external temperature of 115 °C for 1.5 h. Thereafter, the reaction mixture was cooled to room temperature, toluene (150 ml) and water (150 ml) were successively added, and the residue removed by filtration through Celite. The filtrate was washed with toluene (300 ml), the combined organic layer washed with saturated brine (250 ml), dried over anhydrous sodium sulfate and concentrated under reduced pressure. To the residue, a mixture (500 ml) of hexane/ethyl acetate (5/1) was added. The mixture was stirred at room temperature for 10 min, and

then the precipitate removed by filtration. The filtrate was concentrated under reduced pressure, and dried under vacuum at 50 °C, to yield the title compound (3.4 g) as a pale yellow solid (yield 80%). ¹H-NMR (CDCl$_3$, 400 MHz) δ (ppm): 8.17(dd, 1H, J = 8.0, 4.0 Hz, 1H), 7.75 (dd, 1H, J = 8.0, 4.0 Hz), 6.91 (brs, 1H), 6.91 (s, 2H), 6.65 (m, 1H), 3.89 (s, 6H), 3.83 (s, 3H). ¹³C-NMR (CDCl$_3$, 400 MHz) δ (ppm): 153.35, 152.02, 146.49, 140.25, 135.72, 134.05, 115.51, 106.30, 98.34, 60.94, 56.13.

Synthesis of (2E)-3-[2-(3,4,5-trimethoxyanilino)pyridin-3-yl]prop-2-enoic acid t-butyl ester (**3**): In a 100 mL flask, a mixture of **2** (880 mg, 5 mmol), tert-butylacrylate (4 mL, 27.5 mmol), diisopropyl ethylamine (4 mL, 23 mmol), tri-o-tolylphosphine (0.95 g, 3 mmol), Pd$_2$(dba)$_3$ (0.375 g, 0.4 mmol) in anhydrous DMF (20 mL) was stirred at 120 °C (preheated oil bath) for 2 h under nitrogen. After removal of DMF, the residue was purified using column chromatography (0 to 1% DCM:MeOH) to afford 0.95 g of product. ¹H-NMR (CDCl$_3$, 400 MHz) δ (ppm): 8.25 (dd, 1H, J = 8.0, 4.0 Hz), 7.70 (m, 2H), 6.82 (m, 3H), 6.43 (s, 1H), 6.37 (d, 1H, J = 12 Hz), 3.86 (s, 6H), 3.83 (s, 3H), 1.54 (s, 9H). ¹³C-NMR (CDCl$_3$, 400 MHz) δ (ppm): 165.85, 153.36, 153.31, 149.34, 137.48, 136.19, 136.17, 133.98, 123.26, 116.85, 115.52, 98.54, 81.05, 60.91, 56.11, 28.16.

Synthesis of (2E)-3-[2-(3,4,5-trimethoxyanilino)pyridin-3-yl] prop-2-enoic acid (**4**): Ester **3** (0.9 g, mmol) was dissolved in 40% TFA in dichloromethane (50 mL) and stirred at room temperature overnight. The solvent was removed under reduce pressure acetonitrile (3 × 30 mL) and the residue dried under vacuum for 12 h. The resulting solid residue **4** (0.72 g, 93%) was used for coupling with amines without further purification. ¹H-NMR (CDCl$_3$, 400 MHz) δ (ppm): 9.03 (brs, 1H), 8.09 (m, 2H), 7.91 (d, 1H, J = 8.0 Hz), 6.88 (brs, 3H), 6.50 (m, 1H), 3.75 (s, 6H), 3.66 (s, 3H).

Synthesis of (2E)-N-(2-amino-4-chlorophenyl)-3-[2-(3, 4, 5-tri methoxy anilino) pyridin-3-yl] prop-2-enamide (**5**, SH6): In a 100 mL round bottom flask, acid **4** (1 g, 3.03 mmol), DMF (15 mL), HATU (MW-380, 1.4 g, 3.6 mmol) and diisopropyl ethylamine (129.25, 2.10 ml, 12.5 mmol) were stirred together for 15 min. The amine (chlorophenylenediamine, 0.4 g, 2.83 mmol) was added and the solution stirred at 22 °C for 21 h. Thereafter, distilled water (20 mL) was added to precipitate a residue which was removed by vacuum filtration and purified by silica gel chromatography (97:3:: CH$_2$Cl$_2$/MeOH) to afford the desired product as a pale yellow solid. ¹H-NMR (DMSO-D$_6$, 400 MHz) δ (ppm): 9.41 (brs, 1H), 8.54 (s, 1H), 8.19 (d, 1H, J = 4 Hz), 7.87 (m, 2H), 7.41 (d, 1H, J = 8 Hz), 7.00 (s, 2H), 6.84 (m, 3 Hz), 6.61 (dd, 1H, J = 8, 2.4 Hz), 5.30 (brs, 2H), 3.74 (s, 6H), 3.62 (s, 3H).

## The paper explained

### Problem

Liver cancer, particularly hepatocellular carcinoma (HCC), is one of the most aggressive and deadly cancers worldwide, with limited treatment options and poor survival rates. Many cases of liver cancer are driven by proteins that control gene expression, such as transcription factors, which are notoriously difficult to target with drugs. One such transcription factor, SALL4, is linked to liver cancer progression and resistance to existing therapies. While a class of drugs called immunomodulatory imide drugs (IMiDs) can degrade SALL4, they fail to stop SALL4-driven cancers. To overcome this challenge, new drug discovery strategies to directly target SALL4 in liver cancer and other malignancies need to be explored.

### Results

To find a functional new drug targeting SALL4, we analyzed the structures of the SALL4 protein variants SALL4A and SALL4B. It was found that the IMiDs targeting site is not present in the second variant of SALL4, the SALL4B. Next, we crystalized the structure of a different region of SALL4 called zinc finger cluster four (ZFC4), which was present in both variants. Using computer-based molecular docking and phenotypic screens, a promising compound called SH6 was identified. This molecule selectively kills cancer cells that express SALL4 by degrading the protein through CUL4A/CRBN pathway. When tested in mice with tumors derived from human cancer patients, SH6 reduced tumor growth by 87%, showing strong anti-cancer effects. Additionally, SH6 had good properties for drug absorption and stability in the body.

### Impact

We showed that targeting both SALL4A and SALL4B could reduce cancer cell viability and inhibit tumor growth in mouse models. The success of SH6 demonstrates the potential for developing similar drugs for other transcription factors, which are often considered "undruggable." If further clinical studies confirm its effectiveness and safety in humans, SH6 could lead to a new treatment option for liver cancer and other SALL4-driven malignancies, offering hope for cancer patients with few therapeutic choices.

## Data availability

Crystal structure statistic was deposited into the Protein Data Bank (PDB code 8CUC). The mass spectrometry proteomics data have been deposited to the ProteomeXchange Consortium via the PRIDE partner repository with the dataset identifier PXD061151.

The source data of this paper are collected in the following database record: biostudies:S-SCDT-10_1038-S44321-025-00241-3.

## Peer review information

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

## Acknowledgements

This work was supported by the Singapore Ministry of Health's National Medical Research Council (Singapore Translational Research (STaR) Investigator Award STaR18nov-0002 (DGT)); the Singapore Ministry of Education under its Research Centres of Excellence initiative (DGT), National Research Foundation, Singapore, CIRG21nov-0009CIRG21nov-0009 (DGT); NIH/NHLBI P01HL131477-01A1 (DGT); P01 HL158688/HL/NHLBI NIH (LC), the Xiu research fund (LC) and AGA/Jenzabar research fund (LC; GW; and DGT); Breast Cancer Research Foundation BCRF 24-177 (GW), the Northeastern Collaborative Access Team beamlines, funded by the National Institute of General Medical Sciences from the National Institutes of Health (P30 GM124165); the Eiger 16M detector on the 24-ID-E beam line funded by a NIH-ORIP HEI grant (S10OD021527); and resources of the Advanced Photon Source, a U.S. Department of Energy (DOE) Office of Science User Facility operated for the DOE Office of Science by Argonne National Laboratory under Contract No. DE-AC02-06CH11357. SDP acknowledges funding from the Linde Family Foundation, NIBR, 3DC, and DDCF. We thank William Kaelin for the Nluc/Fluc plasmid.

## Author contributions

**Bee Hui Liu**: Conceptualization; Resources; Data curation; Formal analysis; Supervision; Funding acquisition; Validation; Investigation; Methodology; Writing—original draft; Project administration; Writing—review and editing. **Miao Liu**: Data curation; Validation; Investigation. **Sridhar Radhakrishnan**:

Conceptualization; Data curation; Validation; Methodology. **Men-Yuan Dai**: Data curation. **Chaitanya Kumar Jaladanki**: Conceptualization; Data curation. **Chong Gao**: Data curation; Investigation. **Jing Ping Tang**: Data curation. **Kalpana Kumari**: Data curation. **Mei Lin Go**: Conceptualization; Writing—review and editing. **Kim Anh L Vu**: Data curation. **Junsu Kwon**: Data curation. **Hyuk-Soo Seo**: Data curation. **Kijun Song**: Data curation. **Xi Tian**: Data curation. **Li Feng**: Data curation. **Justin L Tan**: Data curation. **Arek V Melkonian**: Investigation. **Zhaoji Liu**: Data curation. **Gerburg Wulf**: Investigation. **Haribabu Arthanari**: Data curation. **Jun Qi**: Conceptualization; Data curation. **Sirano Dhe-Paganon**: Data curation; Validation. **John G Clohessy**: Data curation; Investigation. **Yeu Khai Choong**: Data curation. **J Sivaraman**: Investigation. **Hao Fan**: Data curation; Investigation. **Daniel G Tenen**: Conceptualization; Funding acquisition; Project administration; Writing—review and editing. **Li Chai**: Conceptualization; Resources; Project administration; Writing—review and editing.

Source data underlying figure panels in this paper may have individual authorship assigned. Where available, figure panel/source data authorship is listed in the following database record: biostudies:S-SCDT-10_1038-S44321-025-00241-3.

## Disclosure and competing interests statement

The authors declare no competing interests.

# Expanded View Figures

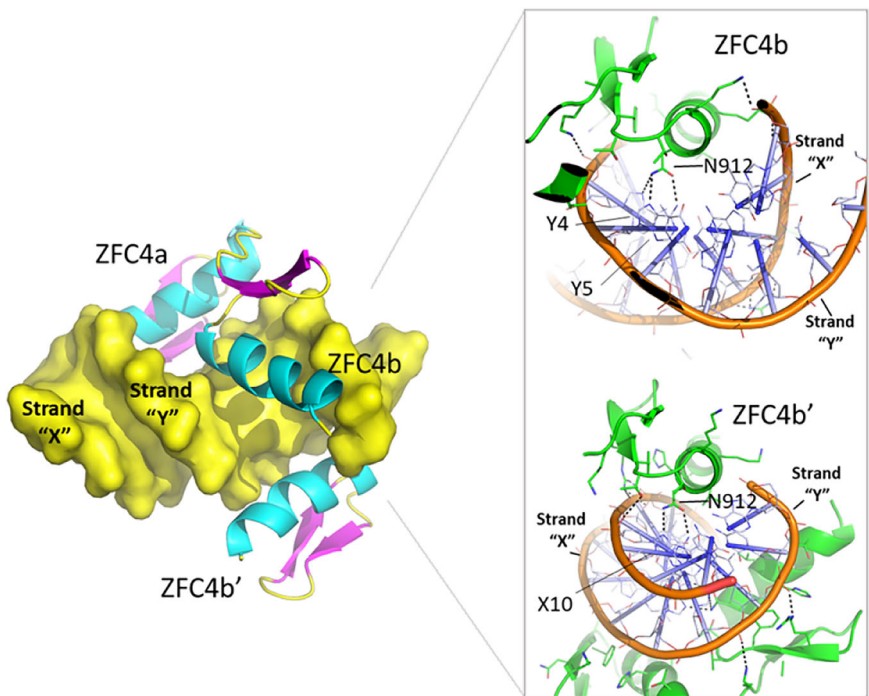

**Figure EV1. Crystal Structure of the fourth SALL4 zinc finger cluster (ZFC4, teal and magenta) bound to DNA (yellow).**

The structural view (left) demonstrates how the two zinc fingers of the first SALL4 molecule (ZFC4b & ZFC4b') and the second zinc finger of the second SALL4 molecule (ZFC4a) in the asymmetric unit bind in the major groove of DNA. Structural view in the box (right) depicting the specific interactions between N912 of SALL4 and adenines at position Y4, Y5, and X10 of DNA.

**A**

SH2                              SH6                              SH7

|  | SH2 | SH6 | SH7 |
|---|---|---|---|
| cLog P [a] | 2.60 | 3.54 | 4.25 |
| PAMPA Effective Permeability Pe ($10^{-6}$cm/s) pH7.4 [b] | - | 66.1 ($\pm$6.8) | 41.2 ($\pm$5.1) |

**B**

|  | Avg P ($10^{6}$cm/s) , pH 7.4 | SD | Classification |
|---|---|---|---|
| Antipyrine | 1.28 | 0.57 | Low |
| Carbamazepine | 65.22 | 0.37 | High |

**Figure EV2. Structures and physicochemical properties of potent analogs SH2, SH6, and SH7.**

(A) These compounds have the lowest EC50 values (<3 μM) on SALL4 high SNU 398 cells. (B) Permeability test (PAMPA) result of the negative control (Antipyrine) and positive control (Carbamazepine).

Linker D: Acrylamide linker is superior to an amide linker.
αβ Unsaturation promotes greater charge delocalization in acrylamide as compared to amide.

Ring C: The ortho diamino groups that constitute the zinc binding motif E are attached to ring C.
Replacing ring C with OH abolishes activity, even though the resulting hydroxamic acid is a zinc binding motif.
Substitution of ring C with X and/or Y (OCH3, Cl, F, CF3, CH3) is permissible but does not have a marked effect on activity.

Ring B: Trisubstitution with sterically bulky methoxy groups favors activity

**Figure EV3. Structural activity of SH library.**

Phenotypic screening of the SH library on SALL4 cell lines (Dataset EV4) revealed three key structure-activity correlations. First, the acrylamide linker D is an indispensable feature for activity. Replacing this linker with an amide resulted in compounds which have markedly lower activity against the SALL4 high cells. In fact, of the compounds with amide linkers (scaffold B-2, Fig. 1H), only four (SH43,47,69,71) have EC50 values ≤20 μM on SALL4 high cells, as compared to 31 out of 35 compounds in acrylamide bearing scaffold B-1. Unlike the amide, electron withdrawal by the carbonyl oxygen in the acrylamide linker delocalizes the positive charge onto the β carbon, resulting in more extensive distribution of charge in the acryan electron deficient β carbon which is susceptible to reaction with electron rich species. It is conceivable that this enhanced reactivity contributed to the greater cell-based potencies of the acrylamide-based SH compounds. A second requirement for potent activity is the ortho diamino motif E derived from the HDAC component in the hybrid scaffold. Of the two ortho-diamino groups, one is embedded within the acrylamide linker D while the other is a substituent on ring C. The ortho positioning of the amino groups is optimal for zinc binding. When Ring C is replaced by hydroxyl (OH) as seen in scaffold B-3 (SH8, SH16, SH24, SH32, SH40), activity was diminished, notwithstanding the retention of the zinc binding motif which is now represented by hydroxamic acid. The diminished activity is likely due to the absence of the sterically larger and bulkier ring C. Lastly, we noted that 6 of the 8 potent compounds (EC50 < 3 μM, all from scaffold B-1) have a trimethoxy substituted ring B. Rings A and B are part of the scaffold found in the antimicrotubule agent E7010, with ring B attached to the ortho position of pyridyl ring A. Mono-substitution of ring B with either methoxy, chloro, fluoro, or trifluoromethyl generally diminished activity. Possibly the sterically bulky and lipophilic trimethoxy substituted ring B has a role in reinforcing interactions with the putative receptor, hence contributing to enhanced potency.

A

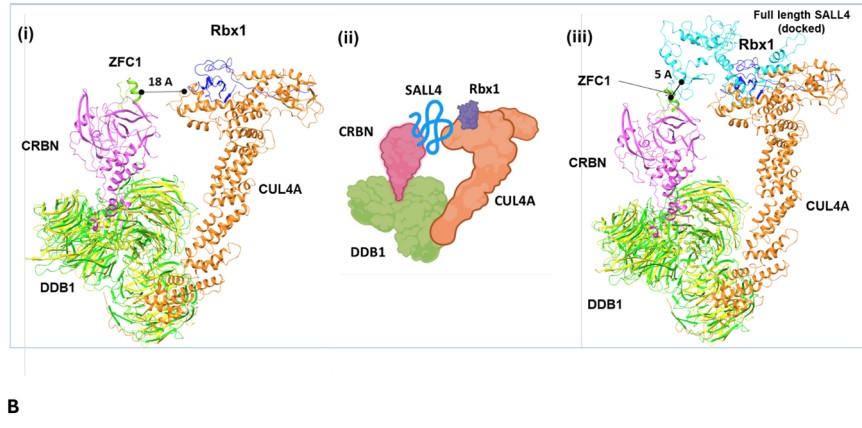

B

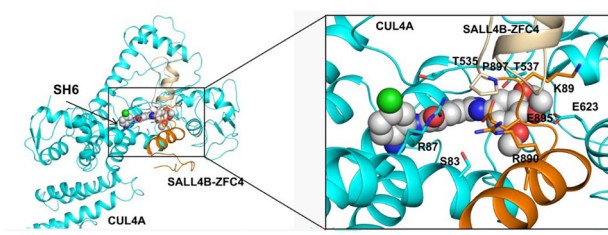

**Figure EV4.   The structural model of SALL4A recruited to the CRBN-DDB1-CUL4A-Rbx1 complex.**

(**A**) ZFC2-4 was predicted to bind to CUL4A near the Rbx1 site. (i) Superimposed 3D structures of the SALL4A ZnF3-CRBN-DDB1(green) complex (PDB ID: 6UML) and DDB1(yellow)-CUL4A-Rbx1 complex (PDB ID: 2HYE) with reference to DDB1. (ii) Cartoon illustration of the SALL4A/B-CRBN-DDB1-CUL4A-RbX1 protein interaction network. (iii) Superimposed 3D structures of the SALL4A ZnF3-CRBN-DDB1(green) complex (PDB ID: 6UML) and DDB1(yellow)-CUL4A-Rbx1-ZFC 2–4 of the SALL4A complex (generated by protein–protein docking) with reference to the DDB1 domain. (**B**) The docking pose of SH6 at the predicted SALL4B ZFC4-CUL4A interface. Wheat and orange = SALL4B ZFC4; Cyan = CUL4A.

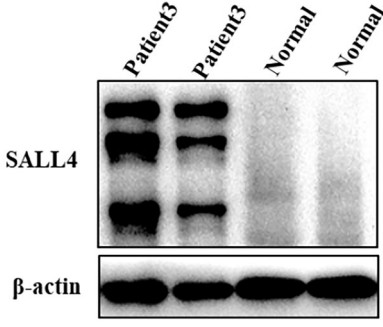

**Figure EV5. Western blot analysis of SALL4 protein expression of patient samples.**

The PDX used in xenograph studies was established from patient 3 with a high level of SALL4 expression. Normal represents adjacent patient sample which did not express detectable levels of SALL4.

