## [Peer Review File · EMBO Molecular Medicine]

Targeting transcription factors through an IMiD independent zinc finger domain

Bee Hui Liu, Miao Liu, Sridhar Radhakrishnan, Men-Yuan Dai, Chaitanya Jaladanki, Chong Gao, Jingping Tang, Kalpana Kumari, Mei Lin Go, Kim Anh Vu, Junsu Kwon, Hyuk-Soo Seo, Kijun Song, Xi Tian, Feng Li, Justin Tan, Arek Melkonian, Zhaoji Liu, Gerburg Wulf, Haribabu Arthanari, Jun Qi, Sirano Dhe-Paganon, John Clohessy, Yeu Khai Choong, J Sivaraman, Hao Fan, Daniel Tenen, and Li Chai

Corresponding author(s): Daniel Tenen (dtenen@bidmc.harvard.edu) , Li Chai (lchai@bwh.harvard.edu)

Review Timeline:

Submission Date:	7th Jan 24
Editorial Decision:	15th Feb 24
Revision Received:	3rd Nov 24
Editorial Decision:	2nd Dec 24
Revision Received:	10th Apr 25
Accepted:	16th Apr 25

Editor: Jingyi Hou

Transaction Report:

15th Feb 2024

Dear Dr. Tenen,

Thank you for the submission of your manuscript to EMBO Molecular Medicine. We have now received feedback from the two reviewers who agreed to evaluate your manuscript. As you will see from the reports below, the referees acknowledge the interest of the study and are overall supportive of your work; however they also comment on multiple aspects of the manuscript that should be strengthened in a revision. In particular, it will be important to include biophysical studies to validate direct binding of SH6 to SALL4. In addition, additional information on experimental setup/methods should be given as pointed out by both reviewers. A justification for how SH6 doses were chosen and the variation in mouse cohort sizes and the different concentrations of SH6 used in the various cellular experiments will need to be provided. Finally it will be important to tone down any conclusions that are not sufficiently supported by the data.

Addressing the reviewers' concerns in full in a point-by-point response will be necessary for further considering the manuscript in our journal, and acceptance of the manuscript will entail a second round of review. EMBO Molecular Medicine encourages a single round of revision only and therefore, acceptance or rejection of the manuscript will depend on the completeness of your responses included in the next, final version of the manuscript. For this reason, and to save you from any frustrations in the end, I would strongly advise against returning an incomplete revision. If you would like to discuss further the points raised by the referees, I am available to do so via email or video. Let me know if you are interested in this option.

We are expecting your revised manuscript within three months, if you anticipate any delay, please contact us. When submitting your revised manuscript, please carefully review the instructions that follow below. We perform an initial quality control of all revised manuscripts before re-review; failure to include requested items will delay the evaluation of your revision.

We require:

4) A .docx formatted letter INCLUDING the reviewers' reports and your detailed point-by-point responses to their comments. As part of the EMBO Press transparent editorial process, the point-by-point response is part of the Review Process File (RPF), which will be published alongside your paper.

5) A complete author checklist, which you can download from our author guidelines (<https://www.embopress.org/page/journal/17574684/authorguide#submissionofrevisions>). Please insert information in the checklist that is also reflected in the manuscript. The completed author checklist will also be part of the RPF.

6) Please note that all corresponding authors are required to supply an ORCID ID for their name upon submission of a revised manuscript.

7) It is mandatory to include a 'Data Availability' section after the Materials and Methods. Before submitting your revision, primary datasets produced in this study need to be deposited in an appropriate public database, and the accession numbers and database listed under 'Data Availability'. Please remember to provide a reviewer password if the datasets are not yet public (see <https://www.embopress.org/page/journal/17574684/authorguide#dataavailability>).

In case you have no data that requires deposition in a public database, please state so in this section. Note that the Data Availability Section is restricted to new primary data that are part of this study. This study includes no data deposited in external repositories.

8) For data quantification: please specify the name of the statistical test used to generate error bars and P values, the number (n) of independent experiments (specify technical or biological replicates) underlying each data point and the test used to calculate p-values in each figure legend. The figure legends should contain a basic description of n, P and the test applied. Graphs must include a description of the bars and the error bars (s.d., s.e.m.). Please provide exact p values.

9) Our journal encourages inclusion of *data citations in the reference list* to directly cite datasets that were re-used and

obtained from public databases. Data citations in the article text are distinct from normal bibliographical citations and should directly link to the database records from which the data can be accessed. In the main text, data citations are formatted as follows: "Data ref: Smith et al, 2001" or "Data ref: NCBI Sequence Read Archive PRJNA342805, 2017". In the Reference list, data citations must be labeled with "[DATASET]". A data reference must provide the database name, accession number/identifiers and a resolvable link to the landing page from which the data can be accessed at the end of the reference. Further instructions are available at .

13) Author contributions: CRediT has replaced the traditional author contributions section because it offers a systematic machine readable author contributions format that allows for more effective research assessment. Please remove the Authors Contributions from the manuscript and use the free text boxes beneath each contributing author's name in our system to add specific details on the author's contribution. More information is available in our guide to authors.

Please also suggest a striking image or visual abstract to illustrate your article as a PNG file 550 px wide x 300-600 px high. Share synopsis text and image, as well as eTOC:

Please note that these would be the final versions and changes during proofing are usually not allowed

16) As part of the EMBO Publications transparent editorial process initiative (see our policy here: https://www.embopress.org/transparent-process#Review_Process), EMBO Molecular Medicine will publish online a Peer Review File (PRF) to accompany accepted manuscripts.

In the event of acceptance, this file will be published in conjunction with your paper and will include the anonymous referee reports, your point-by-point response and all pertinent correspondence relating to the manuscript. Let us know whether you agree with the publication of the PRF and as here, if you want to remove or not any figures from it prior to publication.

I look forward to receiving your revised manuscript.

Yours sincerely,

Poonam Bheda

Poonam Bheda, PhD
Scientific Editor
EMBO Molecular Medicine

***** Reviewer's comments *****

Referee #1 (Comments on Novelty/Model System for Author):

Models applied are of reasonable/good quality. Lack of biophysical studies is noted to validate direct binding of SALL4 degraders.

Referee #1 (Remarks for Author):

The manuscript by Liu et al describes discovery of a new degrader of SALL4 (SH6 compound), which is expected to target ZFC4 domain of both SALL4 isoforms (A and B) and induce SALL4 degradation through CUL4A/CRBN pathway. When tested in SALL4 positive cancer cells, this compound affects cell viability, while deletion of ZFC4 domain abolished its activity in cells. Furthermore, SH6 compound led to a significant reduction in tumor growth in SALL positive xenograft models. Overall, this is an interesting and novel study, which could lead to future development of therapeutically relevant SALL4 degraders. This work, however, suffers from uncomplete mechanistic studies at the molecular level as in vitro binding studies to SALL4 (e.g. ZFC4 domain) and activity in biochemical/biophysical assays have not been conducted for this compound. Since SH6 contains a highly reactive aromatic acrylamide group, it can represent a promiscuous compound and therefore binding and selectivity of this compound are important to be addressed. More specific comments:

- The SH6 compound has been identified through cell-based studies and its direct binding to SALL4/ZFC4 domain has not been validated. It would be critical to validate the molecular mechanism of action of SH6 by performing in vitro binding studies with SALL4 using biophysical approaches and to quantify SH6 binding affinity to the protein.
- The results from Proteomics studies are somewhat confusing and lack complete analysis. For example, why SALL4 was not identified in these studies? Out of many targets degraded (153 proteins), only two zinc finger containing proteins were described (Figure 4B). What are the other targets identified (e.g. the most strongly affected)? Why there is so many degraded proteins? Could these results suggest a promiscuous nature of SH6?
- In line with the above, SH6 compound contains aromatic acrylamide, which represents a highly reactive group that can lead to promiscuous activity in cells by affecting multiple targets. In vitro profiling and in cell selectivity studies (e.g. in a panel of cell lines) could support these studies to validate the target and demonstrate selectivity of this compound.
- The authors state that SH6 binds to both SALL4 and CUL4 to induce degradation of SALL4B, but such statement is not supported by any direct binding experiments (as already pointed out above).
- How the doses for in vivo studies were selected? Very low doses were used (only 1 and 2.5 mg/kg). Why higher doses were not tested to possibly achieve stronger efficacy? The PK indicates relatively low exposure (sub-micromolar) 2-3h after injecting the compound to mice. Is this sufficient then to use such low doses of the compound in efficacy studies?
- What is the half-life of SALL4 in cell/in vivo? This would be important to find out when treating a covalent degrader (such as SH6) in cancer cells with overexpressed SALL4.
- Why the authors used only 4 mice in 2.5mg/kg mice cohort while two other groups contain ~20 mice (Fig. 5C)?
- The concentration of SH6 in various cellular experiments is very different. For example in NanoLuc exp. (Fig. 3C) ~25 uM of SH6 is needed to demonstrate target engagement or in the WB degradation assay 5 uM is required for SALL4 degradation (Fig. 3A), while viability studies show sub-micromolar activity? Why there is such variability in the concentrations of SH6 used in various cellular experiments?

Referee #2 (Remarks for Author):

EMBO-molmed review

Liu et al present an important new approach to discovering drug-like molecules that specifically associate with all isoforms of SALL4 protein and demonstrate that these molecules lead to loss of SALL4 in a tumor cell line. This approach combines structural biology and modelling along with key molecular and cellular data to support the findings. The work is inspired by previous work that showed immunomodulatory imide drugs, such as thalidomide, associate with specific zinc finger domains and

that this association recruits cereblon and its associated ubiquitination machinery, leading to degradation of the protein. SALL4 is an oncofetal protein that is targeted by thalidomide, but only some isoforms can be degraded. SALL4-overexpressing tumor cells are often resistant to this approach and so the authors decided to search for molecules that target the final cluster of zinc fingers (ZFC4) in the protein, which is found in all isoforms and that is essential for DNA binding.

Overall, the data report an exciting new finding of a small molecule directed to ZFC4 that mediates SALL4 degradation. However, not all of the data fully support the authors' interpretation. Moreover, the manuscript lacks clarity to the extent that it confuses, rather than enlightens the reader. With a general toning down of some of the statements and greater clarity in the text, this could be suitable for publication.

Key points:

1. The authors define SALL4 as having four zinc finger clusters, ZFC1-ZFC4, where ZFC1 is a single, atypical zinc finger around residues 72-94, ZFC2 encompasses two zinc fingers (382-432), ZFC3 encompasses three zinc fingers (566-648) and ZFC4 encompasses the two key zinc fingers of interest to this study (870-920). Unfortunately, this set of definitions is at odds with a previous study from the Bird lab where a phylogenetic analysis of the whole SALL protein family showed that there are four zinc finger clusters identifiable across the family and that SALL4 has three of these which are annotated ZFC1 (382-432), ZFC2 (566-648) and ZFC4 (870-920). I would hesitate to call the atypical first SALL4 zinc finger a "cluster" because it is an individual domain, not a cluster of domains. The authors should marry up their definitions with this phylogenetic characterisation of SALL protein family zinc finger clusters.
2. The co-crystal structure of ZFC4 with DNA is barely described. The only overview of the structure is in the supplementary data and it is not clear how this ZFC4 interaction with DNA compares with previous work (Ru et al 2022, Watson et al 2023) because only one side chain interaction with a base is shown, again in the supplementary data, and with a rendering that makes it very difficult to appreciate the interaction. The work should make clear whether and how base recognition is the same in this structure as in other described structures.
3. The authors discuss their observation that while two molecules of SALL4 are present per DNA duplex, only three zinc fingers are visible per duplex, such that final zinc finger, zf8 is observed twice without the corresponding zf7. They provide a highly convoluted explanation as to why this might be, invoking an alternative function for the missing zf7. The explanations for this are likely to be much more mundane and the authors should account for these issues rather than invoking a function for which there is no evidence:
 - a. It is possible that zf7 has been proteolyzed and there is a mix of full length and partial constructs. There is no gel showing protein quality so it is impossible to assess this.
 - b. Zf7 is simply disordered because the DNA duplex used for crystallisation is really short and the arrangement of the domains is a product of the spacing in the crystal, the relative protein concentration and the possible arrangements available.
4. Lots of comparisons are made between the experimental model and various AlphaFold2 models quoting r.m.s.d. scores. The authors need to state whether r.m.s.d. was calculated based on all atoms or main chain atoms or alpha carbons.
5. The description of the binding site search via sitemap is vague, key choices in parameters should be given in the main text rather than the materials and methods.
6. The in-house SH library is only properly described in the supplementary materials. The authors should place some explanation in the main text about what types of molecules are present in this library and why it was used at the point where it is introduced.
7. I'm not sure "surpassed" is an informative descriptor of why the SH library was deemed to be better than the other library. The authors should give a more quantitative/reasoned basis for this.
8. The descriptions of the liver cell lines SNU387 and SNU389 are vague. It is unclear which sets of cell lines are isogenic, apart from expression of SALL4. This should be clarified in the main text.
9. Line 171, section title: "SH6 degrades SALL4..." - the data support that the small molecule SH6 targets SALL4 for degradation via CUL4, whereas the title implies that SH6 directly degrades SALL4.
10. Line 204. It is not clear if this structural modelling has taken into account the likely flexibility of the linkers between zinc finger clusters, which could presumably clash or not clash with parts of CUL4A, depending on how the modelling is carried out. It's hard to discern how this could support an argument that the remaining parts of SALL4 must interact with CUL4A. The authors should clarify.
11. Line 225-226, The authors use modelling to define a site on CUL4A that might also bind SH6 and follow up with CETSA. While the CETSA data show clear stabilisation of SALL4 and proteins likely associated with its degradation, this does not mean that SH6 binds to all of these components; stabilising one component of a larger complex is likely to stabilise other partners. To validate the proposed binding sites of SH6 on SALL4 and CUL4A, the authors need to generate point mutations on both proteins that would prevent SH6 binding and repeat the CETSA experiments.
12. Line 228 "SH6 is a bona fide ZFC4 degrader". The authors somehow redefine "ZFC4" as an entity independent of SALL4. This linguistic switch is unhelpful. For example: "we conducted mass spectrometry analysis on the SH6 degrome to search for additional ZFC4 targets of SH6" is very unclear. This phrasing implied to me that the authors are looking for proteins degraded when targeted by SALL4 ZFC4. However, the experiment is a comparison of SNU398 cell proteomes with and without SH6. These cells express high levels of SALL4, so the experiment can only tell us about SH6-dependent changes, not "ZFC4 targets". This strange wording is carried through into the discussion: "Mass spectrometry analysis of the SH6 degrome searching for additional ZFC4 targets demonstrated that among 294 downregulated proteins, 2 proteins were identified to have ZFC4 sequences". "ZFC4 targets" here means proteins that can be defined as ZFC4s because they are degraded when SH6 is added

and have a similar sequence to SALL4 ZFC4. This is really confusingly stated and must be rephrased.

13. The outcome of the degrome analysis identified two ZBTB proteins that contain zinc finger clusters with "similarity" to SALL4 ZFC4. The authors then state that they can superimpose the structure of ZFC4 on zinc fingers found in ZBTB7A. It is hardly surprising structures can be superposed. This section would be more informative if the authors clearly describe why the ZBTB7A zinc fingers are likely to replicate molecular interactions of ZFC4 of SALL4.

14. The pharmacokinetic profiling of SH6 in mice is interesting. This is done using the SNU398 cells to induce tumors by xenograft. It is not clear if it has previously been shown that similar cells that don't express SALL4 also form tumours? i.e. are the effects seen directly related to SALL4 targeting by SH6?

15. There is a weird justification in the discussion about why the AF2 model, rather than the experimental model was used for the modelling experiments. Benefits of using AF2 are stated as:

a. It "utilizes the entire protein, encompassing regions that may have intrinsic flexibility..." - yet it doesn't seem that the flexibility has been used explicitly in the modelling strategy. The authors need to clarify this point.

b. "within the AlphaFold model, the binding site region is characterized by minimal or no exposure to water. This enhanced structural integrity ... strengthens the predictive ability of our molecular docking" - this argument is absurd. It is trivial to remove water molecules and ligands from a PDB file to make those sites are accessible to docking molecules. Water is apparent in crystal structures when it is highly ordered and it is known that these "structural" water molecules are an intrinsic part of the fold and therefore structural integrity. Proteins function in aqueous environments, so carrying out the modelling without water is convenient but hardly indicative of a physiological event. This whole paragraph should be extensively edited.

16. There appears to be an error in the SALL4 sequence presented in Figure 4C. The sequence of ZFC4 at the first alpha helix is given as SSASALQKHERTH, whereas the UNIPROT sequence, and the sequence deposited by the authors in the PDB is SSASALQIHERTH.

17. Residue numbers should be labelled on Figure 4F and protein names should be given on the image. The legend gives the wrong color key.

18. It would be very helpful if the two zinc fingers of SALL4 ZFC4 were given consistent colouring throughout the figures.

19. This manuscript needs to be extensively edited so that common terms are used consistently, random and unnecessary capitalisations are removed and conventions, such as italicising species names, are applied to an acceptable standard.

Comments from Reviewer #1

Q1: The SH6 compound has been identified through cell-based studies and its direct binding to SALL4/ZFC4 domain has not been validated. It would be critical to validate the molecular mechanism of action of SH6 by performing *in vitro* binding studies with SALL4 using biophysical approaches and to quantify SH6 binding affinity to the protein.

Answer: We thank the reviewer for the suggestions. We first attempted to perform Surface Plasmon Resonance (SPR) and Isothermal Titration Calorimetry (ITC) to measure the binding of SH6 to SALL4 ZFC4. In both assays, the K_d is outside the detection range and could not be established. We then hypothesize that SH6 binds to SALL4 beyond just ZFC4 region, and we have also attempted to make recombinant full-length SALL4 proteins. We have cloned SALL4A and SALL4B using expression plasmids consisting of SALL4A and SALL4B cDNAs (Uniprot Q9UJQ4) with C-terminal TwinStrep tags and overexpressed in 293T cells for purification (detailed protocol and data in annex I). However, the SALL4 proteins were insoluble and tended to aggregate at the molecular weight (MW) of 500kDa to 700kDa, which cannot be used for SPR or ITC experiments. Similar recombinant protein production was attempted with Baculovirus Sf9 cells but to no avail. Due to these technical challenges, we could not demonstrate direct binding of SH6 to SALL4 ZFC4 using these *in vitro* assays.

However, we have obtained evidence in cellular assays that SH6 binding to SALL4 protein requires ZFC4 based on the following results in Fig. 1 below: We have made 293T cells with either SALL4B full-length wild type (SALL4B^{WT}) or a SALL4B mutant lacking ZFC4 (SALL4B^{-ZFC4}). We observed that when treated with SH6, the SALL4B^{-ZFC4} mutant cannot be degraded (A). Next, using a biotin-labelled SH6, we observed that SH6 can pull down SALL4B^{WT} but not SALL4B^{-ZFC4} (B). In addition, when tested using the CETSA assay, SH6 can stabilize SALL4B^{WT}, but not SALL4B^{-ZFC4} (C).

Based on these findings, we have modified our results and conclusion; we now propose that the SH6-mediated SALL4 degradation requires ZFC4, though the precise mechanism remains to be determined.

Q2 The results from Proteomics studies are somewhat confusing and lack complete analysis. For example,

Figure 1. SH6-mediated SALL4B degradation requires ZFC4. A) 293T cells were stably transfected with pFUW-SALL4B wild type (SALL4B^{WT}), or pFUW-SALL4B with deletion of ZFC4 (SALL4B^{-ZFC4}). Cells were treated with SH6, lenalidomide (Len), or DMSO for 24 hours before being harvested for western blot analysis. SH6 mediated-degradation of SALL4B was nearly abolished in the absence of ZFC4, while lenalidomide did not affect SALL4B levels in SALL4B^{-ZFC4} or SALL4B^{WT} cells. B) Co-Immunoprecipitation pull-down assay showing the interaction of SALL4B^{WT} and SH6, but not SALL4B^{-ZFC4} and SH6. 293T SALL4B^{WT} or SALL4B^{-ZFC4} cells were treated with 2.5 μ M biotinylated SH6 for 2 hours, and bound proteins were eluted for western blot analysis. The assays demonstrated that SALL4B was bound and pulled down by biotinylated SH6 in SALL4B^{WT} cells, but not in SALL4B^{-ZFC4} cells. C) 293T SALL4B^{WT} or SALL4B^{-ZFC4} cells were incubated with SH6 for 4 hours prior to Cellular Thermal Shift Assay (CETSA) analysis. Cell lysates were separated by SDS-PAGE and analyzed by Western blotting SALL4. The results demonstrated that SH6 prolonged thermal stability for SALL4B in 293T SALL4B^{WT} cells (i), but not in SALL4B^{-ZFC4} cells (ii), indicating a lack of binding of SH6 to SALL4B without ZFC4.

why SALL4 was not identified in these studies? Out of many targets degraded (153 proteins), only two zinc finger containing proteins were described (Figure 4B). What are the other targets identified (e.g. the most strongly affected)? Why there is so many degraded proteins? Could these results suggest a promiscuous nature of SH6?

Answer: We acknowledge the reviewer's concerns on the proteomic study. To address this, we perform a new set of mass spectrometry analysis using SNU398 cells treated with SH6 and DMSO, with a much lower dose of 2.5 μ M instead of 50 μ M, and labelled the samples with tandem mass tag for higher sensitivity in detection of differentially displayed proteins. Using a cut off of fold change $\leq 5x$, and p value > 0.005 , only 3 proteins were identified as down regulated, with SALL4 amongst these top three down regulated proteins after SH6 treatment (Fig. 2 below). Notably, all these three down regulated proteins are C_2H_2 containing proteins, suggesting that SH6 may preferentially degrade C_2H_2 zinc fingers, and is not a promiscuous degrader. Furthermore, the common IMiD targets, including GSTP1, IKZF, and CK1a, are not affected by SH6, suggesting SH6 differs from IMiDs in terms of their targets. This data is in the main text, lines 275-284, Figure 4D.

Figure 2. Proteomic study on SH6 treated SNU398 cells confirms the downregulation of SALL4 and two additional C_2H_2 containing proteins. A volcano plot from the mass spectrometry data demonstrates the change of the cellular proteins from the SH6 vs DMSO treatment. Down regulated proteins with twofold change and p-value of 0.05 are highlighted in red.

Q3 *In line with the above, SH6 compound contains aromatic acrylamide, which represents a highly reactive group that can lead to promiscuous activity in cells by affecting multiple targets. In vitro profiling and in cell selectivity studies (e.g. in a panel of cell lines) could support these studies to validate the target and demonstrate selectivity of this compound.*

Answer: This is a good point; the acrylamide might target non-selectively in cells. However, our new proteomic study suggested that limited proteins are affected by SH6 treatment (Fig. 2). Furthermore, we tested SH6 in SALL4^{high} vs. SALL4^{low} cells in hepatocellular cancer (HCC) and additional other cancer cell lines below (Fig. 3A (i) and (ii)). It was observed that SH6 selectively affects the viability of SALL4^{high} but not SALL4^{low} cells. We have also tested SH6 in THLE-2 and THLE-3, two non-cancerous liver epithelial cell lines (Fig. 3A (iii)). A similar observation was found that SH6 selectively affects the viability of HCC cells but not liver epithelial cells. To further observe the toxicity effect of SH6, we monitored the overall wellness and body weight of SH6 treated mice in multiple cohorts, including the SNU398 mouse xenograft study in NSG mice (Fig 3B (i)), patient-derived xenograft study in NSG mice (Fig 3B (ii)), and toxicity study in FVB mice (Fig 3B (iii)). Interestingly, besides no body difference with control mice, we found that weight gain was detected in all mouse strains (NSG and FVB) treated with SH6, showing no visible toxicity issues, indicating SH6 does not target promiscuously.

To further examine the likelihood of thiol-Michael adduct formation by the acrylamide moiety in the SH family of compounds, we assessed examine the ¹H NMR spectra of SH4 (a structural analog of SH6 with better solubility) before and after (24h) addition of the nucleophile cysteamine. Here we found no discernible difference in the spectra of SH4 and SH4+cysteamine. Notably, the SH4 trans double bond peaks ($\delta=7.83$ ppm and 6.82 ppm with $J= 16$ Hz) were observed in the SH4+ cysteamine spectrum, suggesting the absence of adduct formation involving the acrylamide residue of SH4. The ¹H NMR data is now described in the main text line 190-197, Figure EV4. Overall, our data supports that SH6 is a selective compound in targeting SALL4-positive cancer cells.

Figure 3. Studies on the selectivity of SH6. A) Selective cytotoxicity effect of SH6 in SALL4 high cell lines in (i) HCC and (ii) additional cancer cell lines, and when compared to untransformed liver epithelial cells THLE2 and THLE3 (iii). B) Mice body weight was measured in three different cohorts: (i) SNU398 mouse xenograft studies in NSG mice; (ii) patient derived xenograft studies in NSG mice; and (iii) toxicity studies in FVB mice. C) ¹H NMR spectrum of the acrylamide double bond. The ¹H NMR spectrum of SH4 was recorded in DMSO-d₆.

Q4 The authors state that SH6 binds to both SALL4 and CUL4 to induce degradation of SALL4B, but such statement is not supported by any direct binding experiments (as already pointed out above).

Answer: We agree with the reviewer. We have performed SPR assays on SH6 treatment of CUL4A recombinant full-length protein. The affinity between SH6 compound and CUL4A was reproducible at 3 different protein immobilization levels: the KD was estimated at 89 μ M, 98 μ M, and 100 μ M for low, medium, and high immobilization levels, respectively (Fig. 4) (Main text line 251-254 Fig. EV6). As we mentioned earlier

in Question 1, we could not get a successful direct binding result for SH6 to SALL4 ZFC4, and subsequently, we attempted to make full-length recombinant SALL4 proteins (SALL4A and SALL4B) for biophysical binding, but to no avail. As explained in Question 1, in the cells, SH6 could not degrade SALL4 in the absence of ZFC4. In addition, SH6 could not pull down or affect the stability of SAL4B mutant protein lacking the ZFC4 domain in the SALL4B^{ZFC4} cells (Fig 1B and Fig1C). Taken together with our previous result that SH6 could not degrade SALL4 in SALL4B^{ZFC4} cells, these new results indicate that SH6 needs ZFC4 to degrade SALL4, albeit the precise mechanism still needs to be defined. We have modified our conclusion and discussion accordingly.

Figure 4. SPR binding data with 1:1 modelling for the interaction between CUL4A and SH6 compound. The same interaction was tested using 3 different immobilization levels for CUL4A: low (1,700 RU) medium (2,800 RU) and high (4,000 RU). Each plot shows a 2-fold dilution series of SH6 compound, at top concentration 300 μM

Q5 How the doses for in vivo studies were selected? Very low doses were used (only 1 and 2.5 mg/kg). Why higher doses were not tested to possibly achieve stronger efficacy? The PK indicates relatively low exposure (sub-micromolar) 2-3h after injecting the compound to mice. Is this sufficient then to use such low doses of the compound in efficacy studies?

Answer: This is a great point. We have since performed dose escalation with a larger cohort mouse xenograft study and found that the mice tolerated 20mg/kg and 40mg/kg well. Both 20mg/kg and 40mg/kg treatment caused significant tumor growth inhibition compared to control tumors, where control (n=20), SH6 20mg/kg (n=20), and SH6 40mg/kg (n=13), had mean tumor volume at 906.9±137mm³, 604.4±53 mm³, and 345.9±63 mm³ respectively (Fig. 5A) (main text line 291-297, Fig 5B(i)&(ii)). In addition, we have created a SALL4-expressing HCC patient-derived xenotransplant (PDX) model to test the therapeutic efficiency of SH6 in treating patient samples. Liver tumor dissected from a 72-year-old male HCC patient with high SALL4 expression were used to generate PDX in mice. The mice were randomly grouped and treated with control (n=13) or 50mg/kg SH6 (n=13). At the study endpoint, SH6 induced a strong therapeutic effect (p<0.0001) with a tumor growth inhibition of 87%, while the mean tumor volume in the control group was 1495±104.5 mm³, compared to 186.3±16.4 mm³ in the SH6-treated group. Furthermore, SALL4 expression is significantly reduced in the SH6-treated PDX tumors (main text 298-313, Fig. EV7, and Fig. 5C).

Figure 5. (A) Mouse xenograft experiments demonstrate a therapeutic potential of SH6 on a SALL4 positive HCC PDX model. 20mg/kg SH6, 40mg/kg SH6, or DMSO were administered every other day to mice transplanted with SU398. Xenografts were harvested at endpoint and measured. (B) Patient Derived Xenograft (PDX) study design. Mice receiving PDX were randomly grouped and treated with vehicle control or 50mg/kg SH6. Tumor harvested at end point was photographed and weighed.

Q6. What is the half-life of SALL4 in cell/in vivo? This would be important to find out when treating a covalent degrader (such as SH6) in cancer cells with overexpressed SALL4.

Answer: To understand the half-life of SALL4 proteins in cells, we performed Cycloheximide chase assay. It was found that endogenous SALL4 proteins have half lives longer than 12hrs, this could allow sufficient time for SH6 to recruit the E3 ligase complex to degrade SALL4 for effective degradation.

Q7. Why the authors used only 4 mice in 2.5mg/kg mice cohort while two other groups contain ~20 mice (Fig. 5C)?

Answer: We would like to thank the reviewer for the comment. In our latest mouse xenograft experiment, we have utilized a much bigger cohort, with n=20 for vehicle control, n=20 for 20mg/kg SH6, and n=13 for 40mg/kg SH6 (main text line 291-297, Fig 5B(i)&(ii)). In our PDX experiment, we have n=13 for vehicle control and n=13 for 50mg/kg SH6 (main text 298-313, Fig. 5C).

Q8. The concentration of SH6 in various cellular experiments is very different. For example in NanoLuc exp. (Fig. 3C) ~25 uM of SH6 is needed to demonstrate target engagement or in the WB degradation assay 5 uM is required for SALL4 degradation (Fig. 3A), while viability studies show sub-micromolar activity? Why there is such variability in the concentrations of SH6 used in various cellular experiments?

Answer: For various cellular experiments, SH6 was used at different dose to account for different cell types, cell number and sensitivity of the assay. Whenever possible, the assays were done with escalating dose with IC₅₀ shown. For example, in the luciferase assay, H1299-SALL4BNLuc cells were seeded at 1000 cells per well in 384 well plates, for luminescence detection after 24hrs incubation. Whereas for the CETSA assays, and cell viability assays, 1x10⁶ per well (SNU398, 293T-SALL4^{WT}, 293T-SALL4^{ZFC4}) and 1500 cells per well (SNU398, SNU387, SNU387-TgSALL4A, SNU387-TgSALL4B, THLE2, THLE3, H661, H838, CAL51, H552, H2030) was seeded respectively.

Comments from Reviewer #2

Q1. *The authors define SALL4 as having four zinc finger clusters, ZFC1-ZFC4, where ZFC1 is a single, atypical zinc finger around residues 72-94, ZFC2 encompasses two zinc fingers (382-432), ZFC3 encompasses three zinc fingers (566-648) and ZFC4 encompasses the two key zinc fingers of interest to this study (870-920). Unfortunately, this set of definitions is at odds with a previous study from the Bird lab where a phylogenetic analysis of the whole SALL protein family showed that there are four zinc finger clusters identifiable across the family and that SALL4 has three of these which are annotated ZFC1 (382-432), ZFC2 (566-648) and ZFC4 (870-920). I would hesitate to call the atypical first SALL4 zinc finger a "cluster" because it is an individual domain, not a cluster of domains. The authors should marry up their definitions with this phylogenetic characterisation of SALL protein family zinc finger clusters.*

Answer: We thank the reviewer for his/her suggestions. We have changed our terminology to match what has been suggested (abstract, main text lines 71,72, 93, 232-267, 345, Fig 1A, Graphical abstract).

Q2. *The co-crystal structure of ZFC4 with DNA is barely described. The only overview of the structure is in the supplementary data and it is not clear how this ZFC4 interaction with DNA compares with previous work (Ru et al 2022, Watson et al 2023) because only one side chain interaction with a base is shown, again in the supplementary data, and with a rendering that makes it very difficult to appreciate the interaction. The work should make clear whether and how base recognition is the same in this structure as in other described structures.*

Answer: We have compared and aligned all three published crystal structures of ZFC4. Although all three structures contained different duplex DNA lengths and crystal packing, all three reveal nearly identical overall conformations. A closer evaluation revealed the same key interactions, including hydrogen bonds between Asn912 and adenosine and a water-mediated interaction between Thr919 and thymidine, supporting a consensus among the structures. (main text line 106-116, Fig 1B(ii)).

Q3. *The authors discuss their observation that while two molecules of SALL4 are present per DNA duplex, only three zinc fingers are visible per duplex, such that final zinc finger, zf8 is observed twice without the corresponding zf7. They provide a highly convoluted explanation as to why this might be, invoking an alternative function for the missing zf7. The explanations for this are likely to be much more mundane and the authors should account for these issues rather than invoking a function for which there is no evidence:*

- a. It is possible that zf7 has been proteolyzed and there is a mix of full length and partial constructs. There is no gel showing protein quality so it is impossible to assess this.
- b. Zf7 is simply disordered because the DNA duplex used for crystallisation is really short and the arrangement of the domains is a product of the spacing in the crystal, the relative protein concentration and the possible arrangements available.

Answer: We thank the reviewer for the feedback here and agree that our preliminary suggestion of an alternative function for the Zf7 was convoluted and premature. After reanalysis, and close inspection of our protein gel and intact mass-spectroscopy which shows no indication of degradation (see Fig 6 below).

Figure 6. Coomassie blue staining and mass spec of recombinant ZFC4 indicating intact ZFC4 was used in crystal study.

We agree with the points the reviewer raised: The second Zf7 is disordered either because the DNA duplex used for crystallization was simply too short and the arrangement of the domains was a product of the spacing in the crystal; or the relative protein concentration and the possible arrangements available; or because Zf8 has a high propensity to bind DNA and as shown by the first molecule might not have extensive sequence specificity and therefore at crystallographically high concentrations of protein will bind to nonspecific adenosine-contain sequences, and may suggest that the observed second zf8 binding mode might be a crystal packing artifact.

To validate our findings, we studied recently published crystal structure of human SALL4 ZFC4 (856–930)(Ru et al, 2022) and murine SALL4 ZFC4 (870–940)(Watson et al, 2023). All three structures were aligned in Fig. 7 below (in our main text, Figure 1B(ii)). Consistent with the report from Ru et al (Ru et al, 2022), our crystallographic findings of human SALL4 ZFC4 (864-929) bound to its consensus DNA binding target revealed an asymmetric unit that contained one molecule of the duplex DNA and two molecules of ZFC4. Both our crystal and the human ZFC4(856-930) crystal(Ru et al, 2022) demonstrated that only one of the two zinc fingers of the second ZFC4 molecule is bound. Furthermore, although all three structures contained different duplex DNA lengths and crystal packing, they reveal nearly identical overall conformations. Additionally, a close-up analysis reveals the same key interactions, namely including hydrogen bonds between Asn912 and adenosine and a water-mediated interaction between Thr919 and thymidine, supporting a consensus among the structures. (main text line 106-117, Figure 1B (ii))

Figure 7. Ligand Interaction Diagram of the SALL4 ZFC4-DNA complex is shown schematically. (i) Key SALL4 residues (N912) are labelled within red circles. Zinc ions are shown as small green circles. The duplex DNA strands are labelled arbitrarily with letters Y and X and numbered sequentially. Key interactions are shown as dotted red arrows. (ii) Alignment of our complex SALL4 structure (light blue) with recently deposited SALL4 complex structures 7Y3I (light orange) and 8A4I (cyan). Coomassie blue staining and mass spec of recombinant ZFC4 indicating intact ZFC4 were used in the crystal study.

Q4. Lots of comparisons are made between the experimental model and various AlphaFold2 models quoting r.m.s.d. scores. The authors need to state whether r.m.s.d. was calculated based on all atoms or main chain atoms or alpha carbons.

Answer: We thank the reviewer for the suggestion. The r.m.s.d. scores were calculated based on the main chain atoms. This is now clarified in main text line 122.

Q5. The description of the binding site search via sitemap is vague, key choices in parameters should be given in the main text rather than the materials and methods.

Answer: We agree. We have now added criteria applied in the binding site search in main text line 123-126 “Docking studies were based on several criteria, including site area, volume, exposure, enclosure, contact properties, hydrophobicity, hydrophilicity, and donor-acceptor ratio. Sites with both a site score and Dscore (druggability) greater than 0.8 were prioritized for detailed analysis.”

Q6. The in-house SH library is only properly described in the supplementary materials. The authors should place some explanation in the main text about what types of molecules are present in this library and why it was used at the point where it is introduced.

Answer: We have since added the description of the SH library in the main text line 140-149 “Structurally, the CSI-SH compounds are hybrid molecules derived from structural fragments of the microtubule inhibitor E7010 and the HDAC inhibitor N-(2-aminophenyl)-3-{4-[(phenylamino)methyl]phenyl}acrylamide, neither of which are known to interact with the SALL4 protein (Fig 1G). The scaffold B-1 is constructed by linking the (N-phenylpyridine-2-amino) moiety (rings A and B) of E7010 to the anilino ring C of the HDAC inhibitor via an acrylamide linker D (Fig 1H). In another iteration of the scaffold, the acrylamide linker is replaced by an amide (B-2). Notable differences between the two linkers are length (acrylamide > amide) and extent of charge delocalization (acrylamide > amide) (Fig 1G). Both scaffolds (B-1, B-2) retain the ortho-diamino group as a zinc binding motif E, but this is replaced by hydroxamic acid (hydroxyl OH in place of ring C) in scaffolds B-3 and B-4 (Fig 1H).”

Q7. I'm not sure "surpassed" is an informative descriptor of why the SH library was deemed to be better than the other library. The authors should give a more quantitative/reasoned basis for this.

Answer: We thank the reviewer for the suggestion. ‘Surpassed’ is now replaced by ‘displayed higher docking scores’ (main text line 133).

Q8. The descriptions of the liver cell lines SNU387 and SNU389 are vague. It is unclear which sets of cell lines are isogenic, apart from expression of SALL4. This should be clarified in the main text.

Answer: Description of the liver cell lines are described as “SNU398, a liver cancer cell line with a high endogenous level of SALL4, and SNU387, a liver cancer cell line with an undetectable level of SALL4 were used for comparison.” (main text line 153-154). We have also added a description of the isogenic cell lines “we

created isogenic cells by over-expressing SALL4A (SNU387-TgSALL4A) or SALL4B (SNU387-TgSALL4B) in SNU387 cells" (main text line 159-161).

Q9. Line 171, section title: "SH6 degrades SALL4..." - the data support that the small molecule SH6 targets SALL4 for degradation via CUL4, whereas the title implies that SH6 directly degrades SALL4.

Answer: This is a good point, we have changed the section title to "SH6 degrades SALL4B through the CUL4A-CRBN pathway" (main text line 199).

Q10. Line 204. It is not clear if this structural modelling has taken into account the likely flexibility of the linkers between zinc finger clusters, which could presumably clash or not clash with parts of CUL4A, depending on how the modelling is carried out. It's hard to discern how this could support an argument that the remaining parts of SALL4 must interact with CUL4A. The authors should clarify.

Answer: We acknowledge that the flexibility of these linkers could result in different conformations, potentially influencing whether the clusters clash with parts of CUL4A. However, the observed clash between SALL4 ZFC1/ZFC4 and CUL4A near the RBX1 domain in our initial model represents one of the probable conformations, suggesting that an interaction might be feasible under certain conditions. To further support this hypothesis, we conducted protein-protein docking analyses using multiple SALL4 fragments (ZFC1-4 of SALL4A and ZFC4 of SALL4B) with CUL4A, followed by a comparison of the resulting complexes with the SALL4-CRL4-CRBN complex. The consistent identification of a potential binding site in multiple superposed and docking-generated models suggests that certain parts of SALL4 might interact with CUL4A.

Q11. Line 225-226, The authors use modelling to define a site on CUL4A that might also bind SH6 and follow up with CETSA. While the CETSA data show clear stabilisation of SALL4 and proteins likely associated with its degradation, this does not mean that SH6 binds to all of these components; stabilising one component of a larger complex is likely to stabilise other partners. To validate the proposed binding sites of SH6 on SALL4 and CUL4A, the authors need to generate point mutations on both proteins that would prevent SH6 binding and repeat the CETSA experiments.

Answer: We agree with the reviewer that the limitation of modeling followed by CETSA on defining an SH6 binding to CUL4A. As also suggested by Reviewer 1, we have established that the binding KD of SH6-CUL4A using Surface Plasmon Resonance (SPR). The affinity between SH6 compound and CUL4A was reproducible at 3 different protein immobilization levels: KD was estimated at 89 μ M; 98 μ M and 100 μ M for low, medium, and high immobilization levels, respectively (main text line 251-254 Fig EV6). However, the binding of SH6 to SALL4 was technically challenging (please see our answers to Reviewer 1). Instead, we have performed CETSA in a SALL4B mutant cell line without ZFC4. It was observed that when SH6 was treated to the cells, it could bind to CUL4A in both SALL4B^{WT} and SALL4B^{ZFC4} overexpressing cells. However, SH6 only binds to SALL4B^{WT} but not SALL4B^{ZFC4} (main text line 270-274 Fig 4C). We realize these results suggest the SH6/SALL4/CUL4A is just one possibility, so we have modified our conclusion and discussion (main text line 373-375).

Q12. Line 228 "SH6 is a bona fide ZFC4 degrader". The authors somehow redefine "ZFC4" as an entity independent of SALL4. This linguistic switch is unhelpful. For example: "we conducted mass spectrometry analysis on the SH6 degrome to search for additional ZFC4 targets of SH6" is very unclear. This phrasing implied to me that the authors are looking for proteins degraded when targeted by SALL4 ZFC4. However, the experiment is a comparison of SNU398 cell proteomes with and without SH6. These cells express high levels of SALL4, so the experiment can only tell us about SH6-dependent changes, not "ZFC4 targets". This strange wording is carried through into the discussion: "Mass spectrometry analysis of the SH6 degrome searching for additional ZFC4 targets demonstrated that among 294 downregulated proteins, 2 proteins were identified to have ZFC4 sequences". "ZFC4 targets" here means proteins that can be defined as ZFC4s because they are degraded when SH6 is added and have a similar sequence to SALL4 ZFC4. This is really confusingly stated and must be rephrased.

Ans: We agree with the reviewer that this section is very confusing, and we have modified this part of the manuscript extensively in the revision. We have since changed the subtitle to 'SH6 mediated SALL4 degradation requires ZFC4' (line 260), and focus on the role of ZFC4 in SH6-mediated SALL4 degradation. We have performed a new set of mass spectrometry analysis using SNU398 treated with SH6 and DMSO, with a much lower dose of 2.5 μ M instead of 50 μ M, and label the samples with tandem mass tag for higher sensitivity in detection of differentially displayed proteins (line 276-285, Fig 4D and Fig 4E).

Q13. The outcome of the degrome analysis identified two ZBTB proteins that contain zinc finger clusters with "similarity" to SALL4 ZFC4. The authors then state that they can superimpose the structure of ZFC4 on zinc fingers found in ZBTB7A. It is hardly surprising structures can be superposed. This section would be more informative if the authors clearly describe why the ZBTB7A zinc fingers are likely to replicate molecular interactions of ZFC4 of SALL4.

Answer: We agree with the reviewer. As we replied to question 12, we decided to take out the degrome concept, and focus on SH6 and SALL4 degradation. We have performed a new set of mass spectrometry analyses using SNU398 treated with SH6 and DMSO, with a much lower dose of 2.5 μM instead of 50 μM , and labeled the samples with a tandem mass tag for higher sensitivity in the detection of differentially displayed proteins. Using a cutoff at fold change $>5x$, and p value > 0.005 , only 3 proteins were identified as down regulated (SALL4, ZFP91, and ZFC653). We have since replaced the old mass spec data with this and described in main text pages 275-284, Figure 4D. Since ZBTB7A was not among the top down-regulated proteins, and in addition we do not have any direct data to demonstrate why the ZBTB7A zinc fingers are likely to replicate molecular interactions of ZFC4 of SALL4, we have deleted ZBTB7A in our revised manuscript.

Q14. The pharmacokinetic profiling of SH6 in mice is interesting. This is done using the SNU398 cells to induce tumors by xenograft. It is not clear if it has previously been shown that similar cells that don't express SALL4 also form tumours? i.e. are the effects seen directly related to SALL4 targeting by SH6?

Answer: As we replied to reviewer 1, We have performed additional mouse xenograft studies using two other doses (20mg/kg and 40mg/kg) for SH6. We found that the mice tolerated 20mg/kg and 40mg/kg doses well, in which there was no significant change of body weight or behavior upon treatment of SH6 throughout the treatment regime. Furthermore, both 20mg/kg and 40mg/kg treatment caused significant tumor growth inhibition, while control tumors (n=20), SH6 20mg/kg (n=20), and SH6 40mg/kg (n=13) tumors demonstrated a mean tumor volume of $906.9 \pm 137 \text{ mm}^3$, $604.4 \pm 53 \text{ mm}^3$, and $345.9 \pm 63 \text{ mm}^3$, respectively (main text line 291-297, Fig 5B(i)&(ii)).

In addition, we have created a SALL4-expressing liver cancer HCC patient-derived xenotransplant (PDX) model to test the therapeutic efficiency of SH6 in treating patient samples. Liver tumor dissected from a 72-year-old male HCC patient with high SALL4 expression were used to generate PDX in mice. The mice were randomly grouped and treated with control (n=13) or 50mg/kg SH6 (n=13). At the study endpoint, SH6 induced a strong therapeutic effect ($p < 0.0001$) with a tumor growth inhibition of 87%, in which the mean tumor volume in the control group was $1495 \pm 104.5 \text{ mm}^3$, compared to $186.3 \pm 16.4 \text{ mm}^3$ in the SH6-treated group.

Furthermore, SALL4 expression is significantly reduced in the SH6-treated PDX tumors (Figure 8 below) (main text 298-313, Fig EV7, and Fig 5C).

Figure 8. Decreased SALL4 expression in SH6-treated PDX tumours. Western blot analysis of SALL4 expression in representative PDX tumors harvested from control-treated (left) and SH6- treated (right) cohorts.

We also monitored the overall wellness and body weight of SH6-treated mice in multiple cohorts, including the SNU398 mouse xenograft study in NSG mice (main text 296-297 Fig 5B (ii)), patient-derived xenograft study in NSG mice (main text 311-313, (Fig 5C (iv) and (v)), and FVB mice (main text 314-316, Fig 5D). Interestingly, beside no difference in body weight was observed in control mice, we found that weight gain was detected in all mouse strains (NSG and FVB) treated with SH6, demonstrating t showing no visible toxicity issues, indicating SH6 does not target promiscuously.

We also tried to use SNU387, the cell line that does not express SALL4 for xenograft experiments. Unfortunately, after months of observation with various cell doses, these cell lines did not form xenograft tumors when we transplanted them into NSG mice.

Q15. *There is a weird justification in the discussion about why the AF2 model, rather than the experimental model was used for the modelling experiments. Benefits of using AF2 are stated as:*

a. It "utilizes the entire protein, encompassing regions that may have intrinsic flexibility..." - yet it doesn't seem that the flexibility has been used explicitly in the modelling strategy. The authors need to clarify this point.

Ans: Thank you for pointing out this mistake. We did not consider flexibility in our docking analysis. We have corrected and modified the statement to: "The AlphaFold model utilizes the entire protein, including regions around the binding site that are not resolved in the ZFC4:DNA crystal structure" (main text line 365-368).

b. "within the AlphaFold model, the binding site region is characterized by minimal or no exposure to water. This enhanced structural integrity ... strengthens the predictive ability of our molecular docking" - this argument is absurd. It is trivial to remove water molecules and ligands from a PDB file to make those sites are accessible to docking molecules. Water is apparent in crystal structures when it is highly ordered and it is known that these "structural" water molecules are an intrinsic part of the fold and therefore structural integrity. Proteins function in aqueous environments, so carrying out the modelling without water is convenient but hardly indicative of a physiological event. This whole paragraph should be extensively edited.

Answer: Thank you for highlighting the lack of clarity in our previous text. Here, by "water" we mean the implicit solvent phase used in the force field for docking simulations, not structural water. We have revised and rewritten this section for more clarity as "Furthermore, the AlphaFold models exhibit a complete binding site with minimal solvent accessible surface area, maintaining a more stable conformation that enhances the predictive accuracy of our molecular docking studies. In contrast, the binding site in the ZFC4:DNA crystal structure is highly exposed, making it more challenging to dock small molecules." (main text line 371-372).

Q16. *There appears to be an error in the SALL4 sequence presented in Figure 4C. The sequence of ZFC4 at the first alpha helix is given as SSASALQKHERTH, whereas the UNIPROT sequence, and the sequence deposited by the authors in the PDB is SSASALQIHERTH.*

Answer: We are sorry for this oversight; we have corrected the above sequence and updated it in the main text, Figure 4E.

Q17. *Residue numbers should be labelled on Figure 4F and protein names should be given on the image. The legend gives the wrong color key.*

Answer: Thank you for your suggestion. Figure 4F is now being replaced by Figure 4E, and color codes are updated.

Q18. *It would be very helpful if the two zinc fingers of SALL4 ZFC4 were given consistent colouring throughout the figures.*

Answer: We appreciate the suggestion. Main text Figure 3F has been updated to have consistent coloring with main text Figures 1D and E.

Q19. *This manuscript needs to be extensively edited so that common terms are used consistently, random and unnecessary capitalisations are removed and conventions, such as italicising species names, are applied to an acceptable standard.*

Ans: We have done detailed and thorough editing in the revision, and hopefully these errors have been corrected..

Annex I.

Cloning and constructs

SALL4A and SALL4B expression plasmids consisted of SALL4A and SALL4B cDNAs (Uniprot Q9UJQ4) with a C-terminal TwinStrep tag (SAWSHPQFEKGGGSGGGSGGSAWSHPQFEK) in pcDNA3.4.2MT (pcDNA3.4 without the NeoR/KanR marker). The plasmid backbone and SALL4 cDNA were amplified via Q5® Hot Start High-Fidelity DNA Polymerase (New England Biolabs) using primers with 15 bp overhangs. Both PCR products were gel purified with the GeneJet Gel Purification Kit (Thermo Fisher) and assembled via In-Fusion cloning (Takara Bio). The K64R mutation was introduced using the QuikChange II XL Site-Directed

Mutagenesis kit (Agilent) to disrupt the nuclear localization signal (NLS) and improve yield.¹ Plasmid DNA was prepared for downstream transfection using the ZymoPURE II Plasmid Maxiprep Kit (Zymo Research).

Cell culture

Expi293 cells (Thermo Fisher) were grown, split, and transfected per the manufacturer's protocol. Briefly, on Day 0, untransfected cells were split to a density of 0.75e6 cells/mL by spinning cells down and resuspending in fresh Expi293 media (Thermo Fisher). A total of 400 mL cell culture was added to a 1L flat bottom vented shake flask. On Day 2, cells were transfected with 400 µg expression plasmid with Expifectamine (Thermo Fisher) per the manufacturer's protocol. On Day 3, the media were supplemented with Enhancer I and Enhancer II from the Expifectamine kit. On Day 7, transfections were harvested by spinning cells at 1000 x *g* for 5 minutes and decanting the supernatant media. Separately, untransfected Expi293 cells were grown to produce denatured Expi293 cell lysate for a downstream purification wash step to remove Hsp70.²

Protein purification

Cell pellets were resuspended in M-PER Extraction Reagent (Thermo Fisher) per the manufacturer's protocol with two modifications: Halt™ Protease Inhibitor Cocktail (Pierce) and an additional 500 mM NaCl were added to the lysis buffer. After lysis was complete, 350 µL BioLock (IBA Biosciences) were added to the solution to prevent binding of biotin present in trace Expi293 media to the Strep-Tactin XT resin. Lysed cell pellets were then spun at 25,000 x *g* for 30 min, after which the clarified lysate was collected in a fresh 50 mL conical vial. Before use, 1 mL Strep-Tactin XT 4Flow resin (IBA Biosciences) was washed three times with TBS (pH 7.4). Washed Strep-Tactin XT resin was added to the lysed supernatant and allowed to rotate continuously at 4 °C for 1 hour, after which the resin was transferred to a chromatography spin column (Pierce). The resin was washed and equilibrated in the following scheme:

1. 8 column volumes (CV) TBS/0.5 mM TCEP
2. 30 min incubation at RT with 100 µM stabilizing dsDNA³
3. 8 CV TBS/0.5 mM TCEP/10 mM ATP/20 mM MgCl₂/300 mM KCl/0.2 mg/mL denatured Expi293 cell lysate
4. 8 CV TBS/0.5 mM TCEP/10 mM ATP/20 mM MgCl₂/300 mM KCl
5. 8 CV TBS/0.5 mM TCEP
6. 20 CV Buffer W/0.5 mM TCEP

The construct was then eluted with 12 CV 1x Buffer BXT (IBA Biosciences). The elution was concentrated on a 30K MWCO Amicon centrifugal concentrator (Millipore), filtered using a 0.22 µm PVDF centrifuge tube filter (Thermo Fisher), and injected onto either a Superdex 200 Increase 10/300 GL or Superose 6 10/300 GL size exclusion column (Cytiva) with 50 mM Tris/100 mM NaCl pH 8.0 as the mobile phase.

Results

Results from a representative preparation of SALL4B are shown below. Fractions of interest during purification were diluted in 4x Laemmli buffer and run on a 4-20% SDS gel, stained with Instant Blue (Abcam). The green arrow indicates SALL4B, whereas the red arrow indicates Hsp70.

pAVM200 SALL4B 041724 001

SDS-PAGE lanes:

1. Wash 2
2. Wash 4
3. Wash 6
4. Wash 8
5. Wash 10
6. Wash 12
7. Wash 14

8. Wash 16
9. Ladder (BioRad Dual Color Standard)
10. Elution 1
11. Elution 2
12. Elution 3
13. Fractions 17-19 (pooled, concentrated on 30K MWCO Amicon)
14. Fraction 26
15. Fraction 32

The apparent molecular weight of the species of interest is ~500-700 kDa which resolves to ~80 kDa on the SDS-PAGE gel, indicating multimerization or aggregation. Similar results were obtained with full-length SALL4A.

References

1. Wu, M.; et al. Identification of the nuclear localization signal of SALL4B, a stem cell transcription factor. *Cell Cycle* **2014**, *13*(9), 1456-1462.
2. Rial, D.V.; Ceccarelli, E.A. Removal of DnaK contamination during fusion protein purifications. *Protein Expression and Purification* **2002**, *25*(3), 503-507.
3. Watson, J.A.; et al. Structure of SALL4 zinc finger domain reveals link between AT-rich DNA binding and Okhiro syndrome. *Life Science Alliance* **2023**, *6*(3), 1-13.

2nd Dec 2024

Dear Dr. Tenen,

Thank you for the submission of your revised manuscript to EMBO Molecular Medicine. Your manuscript has now been re-reviewed by the two original reviewers. Based on their advice (included below), I am pleased to inform you that we will be able to accept your manuscript pending the following final amendments and appropriate response to reviewers:

1) Please download and complete the "Author Checklist", which is required to be published alongside the paper and peer review reports. You can download this file from our author guidelines page:

<https://www.embopress.org/page/journal/17444292/authorguide>

2) We require an ORCID id for corresponding authors - currently Chai Li does not have an ORCID. This has recently been made mandatory for all EMBO Press journals, and is included as a requirement upon submission (for more information please see the authorship guidelines in our guide to authors: <https://www.embopress.org/page/journal/17574684/authorguide>).

3) There are name discrepancies in the manuscript and our submission system. Please ensure that all names match exactly. Currently Kim Anh L. Vu is listed in the manuscript versus Kim Anh Vu Le in our system; Similarly Li Chai is listed in the manuscript versus Chai Li in our system.

4) In the main manuscript file, please include keywords to max. 5.

5) Please remove the Code Availability statement.

6) Mass spectrometry data deposition: Please deposit all mass spectrometry raw data to a suitable community repository (ProteomeXchange member repository for protein mass spectrometry data) and provide the accession codes in the Data Availability statement. The link for the TMT mass spec data available in the Source Data checklist should also be included in the Data Availability statement.

7) Please include a link to the PDB structure in the Data Availability statement.

8) Please rename "Competing Interests" to "Disclosure and competing interests statement". We updated our journal's competing interests policy in January 2022 and request authors to consider both actual and perceived competing interests. Please review the policy <https://www.embopress.org/competing-interests> and update your competing interests if necessary.

9) Author contributions: Please remove it from the manuscript and specify author contributions in our submission system. CRediT has replaced the traditional author contributions section because it offers a systematic machine-readable author contributions format that allows for more effective research assessment. You are encouraged to use the free text boxes beneath each contributing author's name to add specific details on the author's contribution. More information is available in our guide to authors:

<https://www.embopress.org/page/journal/17574684/authorguide#authorshipguidelines>

10) In the Methods, please take care of the following:

- Please rename "Materials and Methods" to "Methods"

- Animals: Please also ensure that housing conditions as well as gender of the animals involved in experiments is reported.

- Cell lines: Please include all information requested in the author checklist for cell lines used in the manuscript (currently clone numbers are missing). Please also be sure to include a sentence in the Methods as to whether or not the cell lines were recently authenticated and tested for mycoplasma contamination and indicate this in the author checklist as well.

- Primers: please ensure qPCR primers used are included in the Methods (or if included in table format, that the table is included in an Appendix file)

- Please ensure that a statement on whether or not blinding was done is included in the Methods even if no blinding was done.

11) All materials and methods need to be described in the main text using our 'Structured Methods' format, which is required for all research articles. According to this format, the Methods section includes a Reagents and Tools Table (listing key reagents, experimental models, software and relevant equipment and including their sources and relevant identifiers) followed by a Methods and Protocols section describing the methods using a step-by-step protocol format. The aim is to facilitate adoption of the methodologies across labs. More information on how to adhere to this format as well as a downloadable template (.docx) for the Reagents and Tools Table can be found in our author guidelines:

<https://www.embopress.org/page/journal/17574684/authorguide#structuredmethods>

12) Please place individual sections of the manuscript in the following order: Title page - Abstract & Keywords - Introduction - Results - Discussion - Methods - Data Availability - Acknowledgements - Disclosure and Competing Interests Statement - The Paper Explained - References - Figure Legends - Expanded View Figure Legends.

13) For the figures and figure legends, please take care of the following:

- Please remove all figures from main manuscript file and leave only main figure legends and EV figure legends placed after the references. Main figures and EV figures should be uploaded as individual, high-resolution files. Please check "Author Guidelines" for more information:

<https://www.embopress.org/page/journal/17574684/authorguide#figureformat>

- You currently have 8 EV figures. We can only accommodate up to 5. The remaining figures can be compiled in an appendix file, with the legends under each figure, and renamed Appendix Figure S1 etc. The appendix should be uploaded in PDF format and needs a table of content with page numbers.

- Figure 5 needs to fit all on a single page.
 - Please make sure to update the callouts of all figures in the main manuscript text (currently figure callouts are missing for Figure 2B, 2D, EV7)
 - Please note that the figure title for figure EV 8 is not provided in the manuscript. This needs to be rectified.
 - Please indicate the statistical test used for data analysis in the legends of figures 3d; 5b, c(iii-vi); EV 8.
 - Please note that the box plots need to be defined in terms of minima, maxima, centre, bounds of box and whiskers, and percentile in the legend of figure 5d.
 - Please note that information related to n is missing in the legends of figures 1f; 3d; 5b, c(iii-vi); 5d.
 - Although 'n' is provided, please describe the nature of entity for 'n' in the legend of figure EV 8.
 - Please note that the error bars are not defined in the legends of figures 5b, c(iii-vi); EV 8.
 - Please note that the red highlighted box is not defined in the legend of figure 3c(i-iv). This needs to be rectified.
- 14) Tables: Please upload all tables as one .xsl file per table and rename them to Dataset EV1-4. Each dataset will need its legend removed from the manuscript and added to the corresponding file in a separate tab. Please update their callouts in main manuscript text.
- 15) References: The two references after Table EV4 need to be incorporated into the main manuscript references or into an Appendix file reference section.
- 16) Funding: Please ensure that all funding sources are entered into the manuscript submission system. Currently the following are missing in our submission system:
- NIH grant P01HL131477-01A1
 - Xiu research fund and AGA/Jenzabar research fund
 - National Institute of General Medical Sciences from the National Institutes of Health (P30 GM124165)
 - NIH-ORIP HEI grant (S10OD021527);
 - Argonne National Laboratory under Contract No. DE-AC02-06CH11357
 - Linde Family Foundation, NIBR, 3DC, and DDCF
- 17) Synopsis:
- Synopsis image: Please remove it from the manuscript and upload it as a high-resolution jpeg, TIFF, or png file 550 pixels wide x (300-600) pixels high.
 - Synopsis text: Please provide a separate file including a short standfirst (maximum of 300 characters, including space), limit the bullet points to max. 5 and upload it as a separate .doc file. Please write the bullet points to summarise the key NEW findings. They should be designed to be complementary to the abstract - i.e. not repeat the same text. We encourage inclusion of key acronyms and quantitative information (maximum of 30 words / bullet point). Please use the passive voice.
 - Please check your synopsis text and image before submission with your revised manuscript. Please be aware that in the proof stage minor corrections only are allowed (e.g., typos).
- 18) Source Data: Please separate out the Source Data files and upload these as a single source data file (zipped) per figure, with the panels clearly visible in the folder structure.
- 19) The B-actin loading controls are missing in the Source Data files for Figure 3A. The B-actin loading control provided in Figure 3D does not match the Source Data file. If you change the figure or the Source Data, please explain the changes that you have made and why. Source Data are missing for Figure 5c(ii)
- 20) The Paper Explained: Please provide "The Paper Explained" and add it to the main manuscript text. Please check "Author Guidelines" for more information. <https://www.embopress.org/page/journal/17574684/authorguide#researcharticleguide>
- 21) As part of the EMBO Publications transparent editorial process initiative (see our policy here: https://www.embopress.org/transparent-process#Review_Process), EMBO Molecular Medicine will publish online a Peer Review File (PRF) to accompany accepted manuscripts. This file will be published in conjunction with your paper and will include the anonymous referee reports, your point-by-point response and all pertinent correspondence relating to the manuscript. Let us know whether you agree with the publication of the PRF and as here, if you want to remove or not any figures from it prior to publication. Please note that the Authors checklist will be published at the end of the PRF.
- 22) Please provide a point-by-point letter INCLUDING my comments as well as the reviewer's reports and your detailed responses (as Word file).

I look forward to reading a new revised version of your manuscript as soon as possible.

Yours sincerely,

Poonam Bheda

Poonam Bheda, PhD
 Scientific Editor
 EMBO Molecular Medicine

***** Reviewer's comments *****

Referee #1 (Comments on Novelty/Model System for Author):

My requests have been properly addressed and I recommend it for publication.

Referee #1 (Remarks for Author):

My requests have been properly addressed and I recommend it for publication.

Referee #2 (Comments on Novelty/Model System for Author):

I do not have the expertise to comment on the patient-derived xenografts.

Referee #2 (Remarks for Author):

The authors have substantially improved the manuscript, removed several parts that were weak or unsubstantiated and significantly improved the data quality and presentation. The majority of my comments have been addressed, although it is disappointing that the authors have still not presented any data to show direct binding of SH6 to SALL4, either through biophysical methods (they show that there are significant technical difficulties that prevent this) or by using point mutations, based on their docking analysis.

The description of the docking has been improved and identifies a pocket of "600 Å²". I'm not sure if this is a typo or something specific to how this pocket defined but I would have expected a volume to have cubic units?

Reviewer's comments

Referee #1 (Comments on Novelty/Model System for Author):

My requests have been properly addressed and I recommend it for publication.

Referee #1 (Remarks for Author):

My requests have been properly addressed and I recommend it for publication.

Referee #2 (Comments on Novelty/Model System for Author):

I do not have the expertise to comment on the patient-derived xenografts.

Referee #2 (Remarks for Author):

The authors have substantially improved the manuscript, removed several parts that were weak or unsubstantiated and significantly improved the data quality and presentation. The majority of my comments have been addressed, although it is disappointing that the authors have still not presented any data to show direct binding of SH6 to SALL4, either through biophysical methods (they show that there are significant technical difficulties that prevent this) or by using point mutations, based on their docking analysis.

The description of the docking has been improved and identifies a pocket of "600 Å²". I'm not sure if this is a typo or something specific to how this pocket defined but I would have expected a volume to have cubic units?

Ans: At line 132-133, the writing was "... a water-filled albeit hydrophobic pocket of approximately 600 Å² formed at the intersection..." the surface area of the pocket were discussed here, not the volume, hence the unit Å² is correct.

Editor's comments

1) Please download and complete the "Author Checklist", which is required to be published alongside the paper and peer review reports. You can download this file from our author guidelines page:

<https://www.embopress.org/page/journal/17444292/authorguide>

2) We require an ORCID id for corresponding authors - currently Chai Li does not have an ORCID. This has recently been made mandatory for all EMBO Press journals, and is included as a requirement upon submission (for more information please see the authorship guidelines in our guide to

authors: <https://www.embopress.org/page/journal/17574684/authorguide>).

Ans: Li Chai's ORCID: 0000-0003-1937-4750

3) There are name discrepancies in the manuscript and our submission system. Please ensure that all names match exactly. Currently Kim Anh L. Vu is listed in the manuscript versus Kim Anh Vu Le in our system; Similarly Li Chai is listed in the manuscript versus Chai Li in our system.

Ans: Author names were changed accordingly.

4) In the main manuscript file, please include keywords to max. 5.

Ans: Five keywords 'SALL4, degrader, IMiD, C2H2, zinc finger cluster four (ZFC4)' have been added on line 45.

5) Please remove the Code Availability statement.

Ans: Code Availability statement removed.

6) Mass spectrometry data deposition: Please deposit all mass spectrometry raw data to a suitable community repository (ProteomeXchange member repository for protein mass spectrometry data) and provide the accession codes in the Data Availability statement. The link for the TMT mass spec data available in the Source Data checklist should also be included in the Data Availability statement.

Ans: 'The mass spectrometry proteomics data have been deposited to the ProteomeXchange Consortium via the PRIDE partner repository with the dataset identifier PXD061151.

<https://www.ebi.ac.uk/pride/archive/projects/PXD061151>' was added to line 691-693

7) Please include a link to the PDB structure in the Data Availability statement.

Ans: Link to PDB added on line 868: <https://www.rcsb.org/structure/8CUC> (line 689-690)

8) Please rename "Competing Interests" to "Disclosure and competing interests statement". We updated our journal's competing interests policy in January 2022 and request authors to consider both actual and perceived competing interests. Please review the policy <https://www.embopress.org/competing-interests> and update your competing interests if necessary.

Ans: "Competing Interests" was changed to "Disclosure and competing interests statement", and updated on line 707-708

9) Author contributions: Please remove it from the manuscript and specify author contributions in our submission system. CRediT has replaced the traditional author contributions section because it offers a systematic machine-readable author contributions format that allows for more effective research assessment. You are encouraged to use the free text boxes beneath each contributing author's name to add specific details on the author's contribution. More information is available in our guide to authors:

<https://www.embopress.org/page/journal/17574684/authorguide#authorshipguidelines>

Ans: Author contribution removed from manuscript.

10) In the Methods, please take care of the following:

- Please rename "Materials and Methods" to "Methods"

Ans: "Materials and Methods" renamed to "Methods" on line 426

- Animals: Please also ensure that housing conditions as well as gender of the animals involved in experiments is reported.

Ans: Animal information including housing conditions added to the manuscript on line 602-604, 611-614, 627-630

- Cell lines: Please include all information requested in the author checklist for cell lines used in the manuscript (currently clone numbers are missing). Please also be sure to include a sentence in the Methods as to whether or not the cell lines were recently authenticated and tested for mycoplasma contamination and indicate this in the author checklist as well.

Ans: Available clone numbers and cell authentication statement added to the manuscript at line 542-551

- Primers: please ensure qPCR primers used are included in the Methods (or if included in table format, that the table is included in an Appendix file)

Ans: In the original text (line 623-624), it was written that 'Part of the tissue was preserved for Western Blot and qPCR analysis to assess SALL4 expression'. However, qPCR was not performed for this manuscript. To avoid confusion, 'and qPCR' was removed from line 624.

- Please ensure that a statement on whether or not blinding was done is included in the Methods even if no blinding was done.

Ans: 'No blinding was done' was added to line 615 and 642.

11) All materials and methods need to be described in the main text using our 'Structured Methods' format, which is required for all research articles. According to this format, the Methods section includes a Reagents and Tools Table (listing key reagents, experimental models, software and relevant equipment and including their sources and relevant identifiers) followed by a Methods and Protocols section describing the methods using a step-by-step protocol format. The aim is to facilitate adoption of the methodologies across labs. More information on how to adhere to this format as well as a downloadable template (.docx) for the Reagents and Tools Table can be found in our author

guidelines: <https://www.embopress.org/page/journal/17574684/authorguide#structuredmethods>

An example of a Method paper with Structured Methods can be found here: <https://www.embopress.org/doi/10.15252/msb.20178071>.

12) Please place individual sections of the manuscript in the following order: Title page - Abstract & Keywords - Introduction - Results - Discussion - Methods - Data Availability - Acknowledgements - Disclosure and Competing Interests Statement - The Paper Explained - References - Figure Legends - Expanded View Figure Legends.

13) For the figures and figure legends, please take care of the following:

- Please remove all figures from main manuscript file and leave only main figure legends and EV figure legends placed after the references. Main figures and EV

figures should be uploaded as individual, high-resolution files. Please check "Author Guidelines" for more information:

<https://www.embopress.org/page/journal/17574684/authorguide#figureformat>

Ans: Figures are removed from main manuscript and figure legends placed after references.

- You currently have 8 EV figures. We can only accommodate up to 5. The remaining figures can be compiled in an appendix file, with the legends under each figure, and renamed Appendix Figure S1 etc. The appendix should be uploaded in PDF format and needs a table of content with page numbers.

Ans: EV4 is now being renamed as Appendix Figure S1 (line 195), and EV6 renamed as Appendix Figure S2 (line 255). EV5 renamed as EV4 (line 232,238,242), EV7 renamed as EV5 (line 303). EV8 renamed as Appendix Figure S3 (line 314).

- Figure 5 needs to fit all on a single page.

Ans: Figure 5 has been fitted on a single page. Figure 5C (i) is now split into Fig 5C (i) and (ii), Fig 5C (ii) is now Fig 5C (iii), Fig 5C (iii) is now Fig 5C (iv) , Fig 5C (iv) is now Fig 5C (v), Fig 5C (v) is now Fig 5C (vi), Fig 5C (vi) is now Fig 5C (vii).

- Please make sure to update the callouts of all figures in the main manuscript text (currently figure callouts are missing for Figure 2B, 2D, EV7)

Ans: Callouts for Figure 2B (line 161), Figure 2D (line 186), and EV7 (currently EV5, line 303) were added to the manuscripts.

- Please note that the figure title for figure EV 8 is not provided in the manuscript. This needs to be rectified.

Ans: EV8 renamed as Appendix Figure S3 (line 314).

- Please indicate the statistical test used for data analysis in the legends of figures 3d; 5b, c(iii-vi); EV 8.

Ans: 'P values were calculated by the two-tailed Student's t-test.' Was added to the legends of figures 3d; 5b, c(iii-vi); EV 8.

- Please note that the box plots need to be defined in terms of minima, maxima, centre, bounds of box and whiskers, and percentile in the legend of figure 5d.

Ans: 'The plot represents the distribution of body weights across SH6 treated mice. The central line within each box corresponds to the median (50th percentile) body weight. The lower and upper bounds of the box represent the 25th (Q1) and 75th (Q3) percentiles, respectively, defining the interquartile range (IQR). Whiskers

extend from the 25th percentile to the minimum value and from the 75th percentile to the maximum value, excluding any outliers. Error bars represent the standard error of the mean (SEM). All data are presented in grams (g). ' was added to the figure legend of Figure 5D

- Please note that information related to n is missing in the legends of figures 1f; 3d; 5b, c(iii-vi); 5d.

'Total number of compounds =676. Shortlisted compounds (n=119) with -5.0 cut off docking scores were highlighted in grey box, n= number of compounds.' were added in legends for Fig 1F. 'Experiments were performed twice with independent cell cultures' were added for Fig 3D legends. '20mg/kg SH6 (n=20), 40mg/kg SH6 (n=13) or vehicle control (n=20) was administered every other day from Day 7 to Day 17.' Was added to the legend for Fig 5B. 'Mice receiving PDX were randomly grouped and treated with vehicle control (n=13) or 50mg/kg SH6 (n=13), n= number of mice. Tumors harvested at end point were photographed C(ii). Four of the representative tumors from each treatment group were analyzed for SALL4 protein expression using western blot.' Tumor growth was measured and traced (n=13 for each group) (iv), and tumor volume measured at endpoint (n=13 for each group).' was added to the legend of Fig 5C. 'Body weight of four healthy FVB/NJ mice receiving 50 mg/kg SH6 daily for 10 days' was added to the legend of Fig 5D.

- Although 'n' is provided, please describe the nature of entity for 'n' in the legend of figure EV 8. 'n= number of mice (n=8)' is now provided in legends of Appendix Figure S3

- Please note that the error bars are not defined in the legends of figures 5b, c(iii-vi); EV 8.

Ans: Error bars = standard error of the mean (SEM) was added to legends of figures 5b, c(iii-vi) and EV8 (currently Appendix Figure S3)

- Please note that the red highlighted box is not defined in the legend of figure 3c(i-iv). This needs to be rectified.

Ans: 'Red box highlighted the difference of the proteins under SH6 or DMSO treatment' was added to the legend for figure 4c(i-iv).

14) Tables: Please upload all tables as one .xsl file per table and rename them to Dataset EV1-4. Each dataset will need its legend removed from the manuscript and added to the corresponding file in a separate tab. Please update their callouts in main manuscript text.

Ans: All tables are now renamed as Dataset EV1-6, and called out in the main manuscript accordingly.

15) References: The two references after Table EV4 need to be incorporated into the main manuscript references or into an Appendix file reference section.

Ans: The two references are incorporated into the Appendix file reference section

16) Funding: Please ensure that all funding sources are entered into the manuscript submission system. Currently the following are missing in our submission system:

- NIH grant P01HL131477-01A1
- Xiu research fund and AGA/Jenzabar research fund
- National Institute of General Medical Sciences from the National Institutes of Health (P30 GM124165)
- NIH-ORIP HEI grant (S10OD021527);
- Argonne National Laboratory under Contract No. DE-AC02-06CH11357
- Linde Family Foundation, NIBR, 3DC, and DDCF

Ans: Funding agencies were updated in the submission systems, except for Xiu research fund and AGA/Jenzabar research fund, and Linde Family Foundation which couldn't be detected by the system. They have been updated in the comment section of the 'Funding'.

17) Synopsis:

- Synopsis image: Please remove it from the manuscript and upload it as a high-resolution jpeg, TIFF, or png file 550 pixels wide x (300-600) pixels high.
- Synopsis text: Please provide a separate file including a short standfirst (maximum of 300 characters, including space), limit the bullet points to max. 5 and upload it as a separate .doc file. Please write the bullet points to summarise the key NEW findings. They should be designed to be complementary to the abstract - i.e. not repeat the same text. We encourage inclusion of key acronyms and quantitative information (maximum of 30 words / bullet point). Please use the passive voice.
- Please check your synopsis text and image before submission with your revised manuscript. Please be aware that in the proof stage minor corrections only are allowed (e.g., typos).

Ans: Synopsis provided as a separate file.

18) Source Data: Please separate out the Source Data files and upload these as a single source data file (zipped) per figure, with the panels clearly visible in the folder structure.

Ans: Source Data files were uploaded as single zip file per figure.

19) The B-actin loading controls are missing in the Source Data files for Figure 3A. The B-actin loading control provided in Figure 3D does not match the Source Data file. If you change the figure or the Source Data, please explain the changes that you have made and why. Source Data are missing for Figure 5c(ii)

Ans: B-actin controls are now updated in the Source Data for Fig 3A. For Fig. 3D, two sets of experiments were performed, the correct set of B-actin loading controls, which matches the source Data file, is now provided in Figure 3D. Source Data for Fig 5C (iii) (previously Fig 5(ii)) is provided.

20) The Paper Explained: Please provide "The Paper Explained" and add it to the main manuscript text. Please check "Author Guidelines" for more information. <https://www.embopress.org/page/journal/17574684/authorguide#researcharticleguide>

21) As part of the EMBO Publications transparent editorial process initiative (see our policy here: https://www.embopress.org/transparent-process#Review_Process), EMBO Molecular Medicine will publish online a Peer Review File (PRF) to accompany accepted manuscripts. This file will be published in conjunction with your paper and will include the anonymous referee reports, your point-by-point response and all pertinent correspondence relating to the manuscript. Let us know whether you agree with the publication of the PRF and as here, if you want to remove or not any figures from it prior to publication. Please note that the Authors checklist will be published at the end of the PRF.

22) Please provide a point-by-point letter INCLUDING my comments as well as the reviewer's reports and your detailed responses (as Word file).

16th Apr 2025

Dear Dr. Tenen,

We are pleased to inform you that your manuscript is accepted for publication and is now being sent to our publisher to be included in the next available issue of EMBO Molecular Medicine.

Yours sincerely,
Jingyi

Jingyi Hou
Senior Editor
EMBO Molecular Medicine
